# A Theoretical Analysis of the Test Error of Finite-Rank Kernel Ridge Regression

**Tin Sum Cheng**    **Aurelien Lucchi**    **Ivan Dokmanić**
Department of Mathematics and Computer Science
University of Basel
{tinsum.cheng, aurelien.lucchi, ivan.dokmanic}@unibas.ch

**Anastasis Kratsios**
Department of Mathematics and Statistics,
McMaster University and Vector Institute,
kratsioa@mcmaster.ca

**David Belius**
Faculty of Mathematics and Computer Science
UniDistance Suisse
david.belius@cantab.ch

## Abstract

Existing statistical learning guarantees for general kernel regressors often yield loose bounds when used with finite-rank kernels. Yet, finite-rank kernels naturally appear in several machine learning problems, e.g. when fine-tuning a pre-trained deep neural network's last layer to adapt it to a novel task when performing transfer learning. We address this gap for finite-rank kernel ridge regression (KRR) by deriving sharp non-asymptotic upper and lower bounds for the KRR test error of any finite-rank KRR. Our bounds are tighter than previously derived bounds on finite-rank KRR, and unlike comparable results, they also remain valid for any regularization parameters.

## 1 Introduction

Generalization is a central theme in statistical learning theory. The recent renewed interest in kernel methods, especially in Kernel Ridge Regression (KRR), is largely due to the fact that deep neural network (DNN) training can be approximated using kernels under appropriate conditions [19, 3, 10], in which the test error is more tractable analytically and thus enjoys stronger theoretical guarantees. However, many prior results have been derived under conditions incompatible with practical settings. For instance [24, 26, 27, 29] give asymptotic bounds on the KRR test error, which requires the input dimension $d$ to tend to infinity. In reality, the input dimension of the data set and the target function is typically finite. A technical difficulty to provide a sharp non-asymptotic bound on the KRR test error comes from the infinite dimensionality of the kernel [1]. While this curse of dimensionality may be unavoidable for infinite-rank kernels (at least without additional restrictions), one may likely derive tighter bounds in a finite-rank setting. Therefore, other works [6, 2] focused on a setting where the kernel is of finite-rank, where non-asymptotic bounds and exact formula of test error can be derived.

Since different generalization behaviours are observed depending on whether the rank of the kernel rank $M$ is smaller than the sample size $N$, one typically differentiates between the under-parameterized ($M < N$) and over-parameterized regime ($M > N$). We focus on the former due to its relevance to several applications in the field of machine learning, including random feature

---

[1]For a fixed input dimension $d$ and sample size $N$, there exists an (infinite-dimensional) subspace in the feature space in which we cannot control the feature vector.

37th Conference on Neural Information Processing Systems (NeurIPS 2023).

models [32, 35, 25, 14, 17] such as reservoir computers [14, 15, 16, 11] where all the hidden layers of a deep learning model are randomly generated, and only the final layer is trainable, or when fine-tuning the final layers of pre-trained deep neural networks for transfer learning [4] or in few-shot learning [38]. The practice of only re-training pre-trained deep neural networks final layer [40, 23] is justified by the fact that earlier layers encode general features which are common to similar tasks thus fine-tuning can support's a network's ability to generalize to new tasks [21]. In this case, the network's frozen hidden layers define a feature map into a finite-dimensional RKHS, which induces a finite-rank kernel; however, similar finite-rank kernels are often considered [13] in which features are extracted directly from several hidden layers in a deep pre-trained neural network which is then fed into a trainable linear regressor.

**Contribution**    Our main objective is to provide sharp non-asymptotic upper and lower bounds for the finite-rank KRR test error in the under-parameterized regime. We make the following contributions:

(i) **Non-vacuous bound in ridgeless case:** In contrast to prior work, our bounds exhibit better accuracy when the ridge parameter $\lambda \to 0$, matching the intuition that a smaller ridge yields a smaller test error;

(ii) **Sharp non-asymptotic lower bound:** We provide a sharp lower bound of test error, which matches our derived upper bound as the sample size $N$ increases. In this sense, our bounds are tight;

(iii) **Empirical validation:** We experimentally validate our results and show our improvement in bounds over [6].

As detailed in Section D, Table 1 compares our results to the available risk-bounds for finite-rank KRR.

Table 1: Comparison of risk-bounds for finite-rank kernel ridge regressors. (*): The bound for the inconsistent case is implicit. See Section 5 for details. (**): By decay, we mean the difference between the upper bound and its limit as $N \to \infty$.

| Assumptions / results | Mohri et al. [30] | Amini et al.[2] | Bach [6] | This paper |
|---|---|---|---|---|
| Include inconsistent case | ✓ | ✗ | (✓)* | ✓ |
| Bias-variance decomposition | ✗ | ✓ | ✓ | ✓ |
| Test error high probability upper bound | ✓ | ✗ | ✓ | ✓ |
| Test error high probability lower bound | ✗ | ✗ | ✗ | ✓ |
| Bounds improve with smaller ridge? | ✗ | - | ✗ | ✓ |
| Upper bound decay rate** | $\sqrt{\frac{\log N}{N}}$ | - | $\lambda + \frac{\log N}{\lambda N}$ | $(\lambda + \frac{1}{N})\sqrt{\frac{\log N}{N}}$ |

**Organization of Paper**    Our paper is organized as follows: Section 2 motivates the importance of the under-parameterized finite-rank KRR. Section 3 introduces the notation and necessary background material required in our results' formulation. Section 4 summarizes our main findings and illustrates their implications via a numerical study. All notation is tabulated in Appendix A for the reader's convenience. All proofs can be found in Appendices B and C, and numerical validation of our theoretical findings in Appendix D.

## 2   Applications

In this section, we motivate the importance of the under-parameterized finite-rank KRR in practical settings. For readers more interested in our main results, please start from Section 3.

**Application: Fine-tuning Pre-Trained Deep Neural Networks For Transfer Learning**    Consider the transfer-learning problem of fine-tuning the final layer of a pre-trained deep neural network model

$f : \mathbb{R}^d \rightarrow \mathbb{R}^D$ so that it can be adapted to a task that is similar to what it was initially trained for. This procedure defines a finite-rank kernel regressor because, for any $x \in \mathbb{R}^d$, $f$ can be factored as

$$f(x) = A\phi(x),$$
$$\phi(x) = \sigma \bullet W^{(L)} \circ \ldots \sigma \bullet W^{(1)}(x),$$

where $A$ is a $D \times d_L$-matrix, $\sigma\bullet$ denotes element-wise application of a univariate non-linear function, for $l = 1, \ldots, L$, $L$ is a positive integer, and $W^{(l)} : \mathbb{R}^{d_{l-1}} \rightarrow \mathbb{R}^{d_l}$, $d = d_0$. In pre-training, all parameters defining the affine maps $W^{(l)}$ in the hidden layers $\sigma \bullet W^{(L)}, \ldots, \sigma \bullet W^{(1)}$ are simultaneously optimized, while in fine-tuning, only the parameters in the final $A$ are trained, and the others are frozen. This reduces $f$ to a finite-rank kernel regressor with finite-rank kernel given for $x, \tilde{x} \in \mathbb{R}^d$ by

$$K(x, \tilde{x}) \stackrel{\text{def.}}{=} \phi(x)^\top \phi(\tilde{x}).$$

Stably optimizing $f$ to the new task requires strict convexity of the KRR problem

$$\min_A \frac{1}{N} \sum_{n=1}^{N} (A\phi(x_n) - y_n)^2 + \lambda \|A\|_F^2, \tag{1}$$

where the hyperparameter $\lambda > 0$ ensures strong convexity of the KRR's loss function and where $\|A\|_F$ denotes the Frobenius norm of A. The unique solution $\hat{A}$ to (1), determines the optimally trained KRR model $\hat{f}(x) \stackrel{\text{def.}}{=} \hat{A}\phi(x)$ corresponding to the finite-rank kernel $K$.

**Application: Random Feature Models**   Popularized by [32] to enable more efficient kernel computations, random feature model has recently seen a substantial spike in popularity and has been the topic of many theoretical works [20, 27]. Note that random feature models are finite-rank kernel regressors once their features are randomly sampled [32, 35, 25, 14, 11].

**Application: General Use**   We emphasize that, though fine-tuning and random feature models provide simple typical examples of when finite-rank KRR arise in machine learning, there are several other instances where our results apply, e.g. when deriving generalization bounds for infinite-rank kernels by truncation, thereby replacing them by finite-rank kernels; e.g. [27].

## 3   Preliminary

We now formally define all concepts and notation used throughout this paper. A complete glossary is found in Appendix A.

### 3.1   The Training and Testing Data

We fix a (non-empty) input space $\mathcal{X} \subset \mathbb{R}^d$ and a *target function* $\tilde{f} : \mathcal{X} \rightarrow \mathbb{R}$ which is to be learned by the KRR from a finite number of i.i.d. (possibly noisy) samples $\mathbf{Z} \stackrel{\text{def.}}{=} (\mathbf{X}, \mathbf{y}) = \left((x_i)_{i=1}^N, (y_i)_{i=1}^N\right) \in \mathbb{R}^{d \times N} \times \mathbb{R}^N$. The inputs $x_i$ are drawn from a *sampling distribution* $\rho$ on $\mathcal{X}$ and outputs are modelled as $y_i = \tilde{f}(x_i) + \epsilon_i \in \mathbb{R}^{N \times 1}$ for some i.i.d. independent random variable $\epsilon_i$ which is also independent of the $\mathbf{X}$, satisfying $\mathbb{E}[\epsilon] = 0$ and $\mathbb{E}[\epsilon^2] \stackrel{\text{def.}}{=} \sigma^2 \geq 0$. Our analysis is set in the space of all square-integrable "function" with respect to the sampling distribution $L_2(\rho) \stackrel{\text{def.}}{=} \{f : \mathcal{X} \rightarrow \mathbb{R} : \mathbb{E}_{x \sim \rho}[f(x)^2] < \infty\}$.

We abbreviate $\mathbf{y} = \tilde{f}(\mathbf{X}) + \boldsymbol{\epsilon} \in \mathbb{R}^{N \times 1}$, where $\tilde{f}(\mathbf{X}) = [\tilde{f}(x_i)]_{i=1}^N$ and $\boldsymbol{\epsilon} = [\epsilon_i]_{i=1}^N$.

### 3.2   The Assumption: The Finite-Rank Kernel Ridge Regressor

As in [2, 6], we fix a rank $M \in \mathbb{N}_+$ for the kernel $K$.

**Definition 3.1** (Finite Rank Kernel). *Let $M$ be a positive integer. A (finite) rank-$M$ kernel $K$ is a map $K : \mathcal{X} \times \mathcal{X} \rightarrow \mathbb{R}$ defined for any $x, x' \in \mathcal{X}$ by*

$$K(x, x') = \sum_{k=1}^{M} \lambda_k \psi_k(x) \psi_k(x'), \tag{2}$$

*where the positive numbers $\{\lambda_k\}_{k=1}^M$ are called the eigenvalues, and orthonormal functions $\psi_k \in L_\rho^2$ are called the eigenfunctions.* [2]

**Remark 3.2** (Eigenvalues are Ordered). *Without loss of generality, we will assume that the kernel $K$'s eigenvalues are ordered $\lambda_1 \geq \lambda_2 \geq \cdots \geq \lambda_M > 0$.*

We denote by $\mathcal{H} \stackrel{\text{def.}}{=} \mathrm{span}\{\lambda_1^{-1/2}\psi_1, \ldots, \lambda_M^{-1/2}\psi_M\}$ the reproducing kernel Hilbert space (RKHS) associated to $K$. See Appendix B for additional details on kernels and RKHSs.

Together, the kernel $K$ and a ridge $\lambda > 0$ define an optimal regressor for the training data $\mathbf{Z}$.

**Definition 3.3** (Kernel Ridge Regressor (KRR)). *Fix a ridge $\lambda > 0$, the regressor $f_{\mathbf{Z},\lambda}$ of the finite-rank kernel $K$ is the (unique) minimizer of*

$$f_{\mathbf{Z},\lambda} \stackrel{\text{def.}}{=} \min_{f \in \mathcal{H}} \frac{1}{N} \sum_{i=1}^N (f(x_i) - y_i)^2 + \lambda \|f\|_{\mathcal{H}}^2. \tag{3}$$

### 3.3 The Target Function

The only regularity assumed of the target function $\tilde{f}$ is that it is square-integrable with respect to the sampling distribution $\rho$; i.e. $\tilde{f} \in L_\rho^2$. We typically assume that $\tilde{f}$ contains strictly more features than can be expressed by the finite-rank kernel $K$. Thus, $\tilde{f}$ can be arbitrarily complicated even if the kernel $K$ is not. Since $\{\psi_k\}_{k=1}^M$ is an orthonormal set of $L_\rho^2$, we decompose the target function $\tilde{f}$ as

$$\tilde{f} = \underbrace{\sum_{k=1}^M \tilde{\gamma}_k \psi_k}_{\tilde{f}_{\leq M}} + \tilde{\gamma}_{>M} \psi_{>M}, \tag{4}$$

for some real numbers $\tilde{\gamma}_k$'s and $\tilde{\gamma}_{>M}$ and for some normal function $\psi_{>M}$ orthogonal to $\{\psi_k\}_{k=1}^M$. We call $\psi_{>M}$ the orthonormal complement and $\tilde{\gamma}_{>M}$ the complementary coefficient. The component $\tilde{f}_{\leq M} \stackrel{\text{def.}}{=} \sum_{k=1}^M \tilde{\gamma}_k \psi_k$ of $\tilde{f}$ is in $\mathcal{H}$. For the case $\tilde{\gamma}_{>M} = 0$, we call it a *consistent* case, as the target function $\tilde{f}$ lies in the hypothesis set $\mathcal{H}$, else we call it an *inconsistent* case.

Alternatively, the orthonormal complement $\psi_{>M}$ can be understood as some input-dependent noise. Assume we have chosen a suitable finite rank kernel $K$ with corresponding RKHS $\mathcal{H}$ such that the target function lies in $\mathcal{H}$. For this purpose, we can write the target function as $\tilde{f}_{\leq M} = \sum_{k=1}^M \tilde{\gamma}_k \psi_k$ for some real numbers $\tilde{\gamma}_k$'s. Suppose that we sample in a noisy environment; then for each sample input $x_i$, the sample output $y_i$ can be written as

$$y_i = \underbrace{\tilde{f}_{\leq M}(x_i)}_{\text{true label}} + \underbrace{\tilde{\gamma}_{>M} \psi_{>M}(x_i)}_{\text{input-dependent noise}} + \underbrace{\epsilon_i}_{\text{input-independent noise}}. \tag{5}$$

### 3.4 Test Error

Next, we introduce the subject of interest of this paper in more detail. Our statistical analysis quantifies the deviation of the learned function from the ground truth of the *test error*.

**Definition 3.4** (KRR Test Error). *Fix a sample $\mathbf{Z} = (\mathbf{X}, \tilde{f}(\mathbf{X}) + \epsilon)$. The finite-rank KRR (3)'s test error is*

$$\mathcal{R}_{\mathbf{Z},\lambda} \stackrel{\text{def.}}{=} \mathbb{E}_{x,\epsilon}\left[(f_{\mathbf{Z},\lambda}(x) - \tilde{f}(x))^2\right] = \mathbb{E}_\epsilon\left[\int_{\mathcal{X}} \left(f_{\mathbf{Z},\lambda}(x) - \tilde{f}(x)\right)^2 d\rho(x)\right]. \tag{6}$$

*The analysis of this paper also follows the classical bias-variance decomposition, thus we write*

$$\mathcal{R}_{\mathbf{Z},\lambda} = bias + variance,$$

*where bias measures the error when there is no noise in the label, that is,*

$$bias \stackrel{\text{def.}}{=} \int_{\mathcal{X}} \left(f_{(\mathbf{X}, \tilde{f}(\mathbf{X})),\lambda}(x) - \tilde{f}(x)\right)^2 d\rho(x), \tag{7}$$

*and variance is defined to be the difference between test error and bias: variance $\stackrel{\text{def.}}{=} \mathcal{R}_{\mathbf{Z},\lambda} - bias$.*

---

[2]This means that $\int_{\mathcal{X}} \psi_k(x)\psi_l(x)d\rho(x) = \delta_{kl}$ for all $1 \leq k \leq l \leq M$.

# 4 Main Result

We now present the informal version of our main result, which gives high-probability upper and lower bounds on the test error. This result is obtained by bounding both the bias and variance, and the probability that the bounds hold is quantified with respect to the sampling distribution $\rho$. Here we emphasize the condition that $N > M$ and hence our statement is valid only in under-parametrized case.

We can assume the data-generating distribution $\rho$ and eigenfunctions $\psi_k$ are well-behaved in the sense that:

**Assumption 4.1** (Sub-Gaussian-ness). *We assume that the probability distribution of the random variable $\psi_k(x)$, where $x \in \rho$, has sub-Gaussian norm bounded by a positive constant $G > 0$, for all $k \in \{1, ..., M\} \cup \{> M\}$[3].*

In particular, if the random variable $\psi_k(x)$ is bounded, the assumption 4.1 is fulfilled.

**Theorem 4.2** (High Probability Bounds on Bias and Variance). *Suppose Assumption 4.1 holds, for $N > M$ sufficient large, there exists some constants $C_1, C_2$ independent to $N$ and $\lambda$ such that, with a probability of at least $1 - 2/N$ w.r.t. random sampling, we have the following results simultaneously:*

*(i)* ***Upper-Bound on Bias:*** *The bias is upper bounded by:*

$$bias \leq \tilde{\gamma}_{>M}^2 + \lambda \|\tilde{f}_{\leq M}\|_{\mathcal{H}}^2 + \left(\frac{1}{4}\|\tilde{f}\|_{L_\rho^2}^2 + 2\lambda\|\tilde{f}_{\leq M}\|_{\mathcal{H}}^2\right)\sqrt{\frac{\log N}{N}} + C_1 \frac{\log N}{N}, \quad (8)$$

*where we denote $\tilde{f}_{\leq M} \overset{\text{def.}}{=} \sum_{k=1}^M \tilde{\gamma}_k \psi_k = \tilde{f} - \tilde{\gamma}_{>M}\psi_{>M}$;*

*(ii)* ***Lower-Bound on Bias:*** *The bias is lower bounded by an analogous result:*

$$bias \geq \tilde{\gamma}_{>M}^2 + \frac{\lambda^2 \lambda_M}{(\lambda_M + \lambda)^2}\|\tilde{f}_{\leq M}\|_{\mathcal{H}}^2 - \left(\frac{1}{4}\|\tilde{f}\|_{L_\rho^2}^2 + \frac{2\lambda^2}{\lambda_1 + \lambda}\|\tilde{f}_{\leq M}\|_{\mathcal{H}}^2\right)\sqrt{\frac{\log N}{N}} - C_1 \frac{\log N}{N}. \quad (9)$$

*(iii)* ***Upper-Bound on Variance:*** *The variance is upper bounded by:*

$$variance \leq \sigma^2 \frac{M}{N}\left(1 + \sqrt{\frac{\log N}{N}} + C_2 \frac{\log N}{N}\right); \quad (10)$$

*(iv)* ***Lower-Bound on Variance:*** *The variance is lower bounded by an analogous result:*

$$variance \geq \frac{\lambda_M^2}{(\lambda_M + \lambda)^2}\sigma^2 \frac{M}{N}\left(1 - \sqrt{\frac{\log N}{N}}\right) - C_2 \sigma^2 \frac{M}{N}\frac{\log N}{N}. \quad (11)$$

*For $\lambda \to 0$, we have a simpler bound on the bias: with a probability of at least $1 - 2/N$, we have*

$$\lim_{\lambda \to 0} bias \leq \tilde{\gamma}_{>M}^2\left(1 + \frac{\log N}{N}\right) + 6\tilde{\gamma}_{>M}^2\left(\frac{\log N}{N}\right)^{\frac{3}{2}};$$
$$\lim_{\lambda \to 0} bias \geq \tilde{\gamma}_{>M}^2\left(1 - \frac{\log N}{N}\right) - 6\tilde{\gamma}_{>M}^2\left(\frac{\log N}{N}\right)^{\frac{3}{2}}. \quad (12)$$

*If $\tilde{\gamma}_{>M}^2 = 0$, then we are in the consistent case, meaning that $\tilde{f}$ belongs to $\mathcal{H}$. In this case, we have a simpler bound on the bias: with a probability of at least $1 - 2/N$, we have*

$$bias \leq \lambda\|\tilde{f}\|_{\mathcal{H}}^2\left(1 + 2\sqrt{\frac{\log N}{N}}\right) + C_1 \frac{\log N}{N}. \quad (13)$$

---

[3]it means the orthonormal complement $\psi_{>M}$ is also mentioned in the assumption.

*Proof Sketch:* The main technical tools are 1) more careful algebraic manipulations when dealing with terms involving the regularizer $\lambda$ and 2) the use of a concentration result for a sub-Gaussian random covariance matrix in [37] followed by the Neumann series expansion of a matrix inverse. Hence, unlike previous work, our result holds for any $\lambda$, which can be chosen independence to $N$. The complete proof of this result, together with the explicit form of the lower bound, is presented in Theorems C.19 and C.20 in the Appendix C. $\qquad\square$

**Remark 4.3.** *Note that one can also easily derive bounds on the expected value of the test error to allow comparisons with prior works, such as [6, 36, 33].*

**Remark 4.4.** *The main technical difference from prior works is that we consider the basis $\{\psi_k\}_{k=1}^M$ in the function space $L_\rho^2$ instead of $\{\lambda_k^{-1/2}\psi_k\}_{k=1}^M$ in the RKHS $\mathcal{H}$. This way, we exploit decouple the effect of spectrum from the sampling randomness to obtain a sharper bound. For further details, see Remark C.12 in Appendix C.*

Combining Theorem 4.2 and the bias-variance decomposition in Definition 3.4, we have both the upper and lower bounds of the test error on KRR.

**Corollary 4.4.1.** *Under mild conditions on the kernel $K$, for $N$ sufficiently large, there exist some constants $C_1, C_2$ independent to $N$ and $\lambda$ such that, with a probability of at least $1 - 2/N$ w.r.t. random sampling, we have the bounds on the test error $\mathcal{R}_{\mathbf{Z},\lambda}$:*

$$\mathcal{R}_{\mathbf{Z},\lambda} \leq \tilde{\gamma}_{>M}^2 + \lambda\|\tilde{f}_{\leq M}\|_{\mathcal{H}}^2 + \left(\frac{1}{4}\|\tilde{f}\|_{L_\rho^2}^2 + 2\lambda\|\tilde{f}_{\leq M}\|_{\mathcal{H}}^2\right)\sqrt{\frac{\log N}{N}} + C_1\frac{\log N}{N}$$
$$+ \sigma^2\frac{M}{N}\left(1 + \sqrt{\frac{\log N}{N}} + C_2\frac{\log N}{N}\right);$$

$$\mathcal{R}_{\mathbf{Z},\lambda} \geq \tilde{\gamma}_{>M}^2 + \frac{\lambda^2\lambda_M}{(\lambda_k+\lambda)^2}\|\tilde{f}_{\leq M}\|_{\mathcal{H}}^2 - \left(\frac{1}{4}\|\tilde{f}\|_{L_\rho^2}^2 + \frac{2\lambda^2}{\lambda_1+\lambda}\|\tilde{f}_{\leq M}\|_{\mathcal{H}}^2\right)\sqrt{\frac{\log N}{N}} - C_1\frac{\log N}{N}$$
$$+ \sigma^2\frac{M}{N}\left(1 - \sqrt{\frac{\log N}{N}} - C_2\frac{\log N}{N}\right).$$

*In particular, we have:*

$$\lim_{\lambda\to 0}\mathcal{R}_{\mathbf{Z},\lambda} \leq \left(1 + \frac{\log N}{N}\right)\tilde{\gamma}_{>M}^2 + 6\tilde{\gamma}_{>M}^2\left(\frac{\log N}{N}\right)^{\frac{3}{2}} + \sigma^2\frac{M}{N}\left(1 + \sqrt{\frac{\log N}{N}} + C_2\frac{\log N}{N}\right).$$

*The corresponding lower bounds are given analogously:*

$$\lim_{\lambda\to 0}\mathcal{R}_{\mathbf{Z},\lambda} \geq \left(1 - \frac{\log N}{N}\right)\tilde{\gamma}_{>M}^2 - 6\tilde{\gamma}_{>M}^2\left(\frac{\log N}{N}\right)^{\frac{3}{2}} + \sigma^2\frac{M}{N}\left(1 - \sqrt{\frac{\log N}{N}} - C_2\frac{\log N}{N}\right).$$

See Section 5 for more details and validations of Theorem 4.2.

# 5 Discussion

In this section, we first elaborate on the result from Theorem 4.2, which we then compare in detail to prior works, showcasing the improvements this paper makes. Finally, we discuss several future research directions.

## 5.1 Elaboration on Main Result

**Bias** From Eq. (8), we can draw the following observations: 1) The term $\tilde{\gamma}_{>M}^2$ in the upper bound is the finite rank error due to the inconsistency between RKHS and the orthogonal complement of the target function, which cannot be improved no matter what sample size we have. We can also view this term as the sample-dependent noise variance (see Eq. (5)). Hence, unlike the sample-independent noise variance $\sigma$ in Eq. (10), the sample-dependent noise variance does not vanish when $N \to \infty$.

2) Note that the third term $\frac{1}{4}\|\tilde{f}\|_{L_\rho^2}^2\sqrt{\frac{\log N}{N}}$ is a residue term proportional to $\|\tilde{f}\|_{L_\rho^2}^2$ and vanishes

when $N \to \infty$, which means we have better control of the sample-dependent noise around its expected value for large $N$. Also, note that the factor $\|\tilde{f}\|^2_{L^2_\rho} = \sum_{k=1}^M \tilde{\gamma}_k^2 + \tilde{\gamma}_{>M}^2$ depends solely on the target function $\tilde{f}$ but not on the kernel $K$ or ridge parameter $\lambda$. 3) The second plus the fourth terms $\left(1 + 2\sqrt{\frac{\log N}{N}}\right) \lambda \|\tilde{f}_{\leq M}\|^2_\mathcal{H}$ depends strongly on the kernel training: the sum is proportional to the ridge $\lambda$, and the RKHS norm square $\|\tilde{f}_{\leq M}\|^2_\mathcal{H} = \sum_{k=1}^M \frac{\tilde{\gamma}_k^2}{\lambda_k}$ measures how well-aligned the target functions with the chosen kernel $K$ is. Again, a larger $N$ favors the control as the residue $\sqrt{\frac{\log N}{N}} \to 0$ as $N \to \infty$. 4) The fifth term $C_1 \frac{\log N}{N}$ is a small residue term with fast decay rate, which the other terms will overshadow as $N \to \infty$.

**Ridge parameter** The bounds in Eq. (12) demonstrate that in the ridgeless case, the bias can be controlled solely by the finite rank error $\tilde{\gamma}_k^2$ with a confidence level depending on $N$; also the upper and lower bounds coincide as $N \to \infty$.

**Variance** For the variance bounds in Eq. (10), we have similar results: the variance can be controlled solely by the (sample-independent) noise variance $\sigma^2$ with a confidence level depending on $N$, also the upper and lower bounds coincides as $N \to \infty$.

## 5.2 Comparison with Prior Works

We discuss how our results add to, and improve on, what is known about finite-rank KRRs following the presentation in Table 1.

**Classical tools to study generalization** The test error measures the average difference between the trained model and the target function. It is one of the most common measures to study the generalization performance of a machine learning model. Nevertheless, generally applicable statistical learning-theoretic tools from VC-theory [8], (local) Rademacher complexities [22, 7, 30], PAC-Bayes methods [1, 28], or smooth optimal transport theory [18] all yield pessimistic bounds on the KRR test error. For example, Mohri et al. [30] bound the generalization error using Rademacher Complexity: there exists some constant $C > 0$ independent to $N$ and $\lambda$ such that, with a probability of at least $1 - \tau$:

$$\text{test error} - \text{train error} \leq \frac{C}{\sqrt{N}} \left(1 + \frac{1}{2}\sqrt{\frac{\log \frac{1}{\tau}}{2}}\right). \tag{14}$$

If we set $\tau$ to $2/N$, the decay in the generalization gap is $\mathcal{O}\left(\sqrt{\frac{\log N}{N}}\right)$, which is too slow compared to other kernel-specific analyses.

**Truncated KRR** Amini et al. [2] suggests an interesting type of finite-rank kernels: for any (infinite-rank) kernel $K^{(\infty)}$, fix a sample $\mathbf{Z} = (\mathbf{X}, \mathbf{y})$ of size $N$ and define a rank-$N$ kernel $K$ by the eigendecomposition of the kernel matrix $\mathbf{K}^{(\infty)}$. Note that different random sample results in a completely different rank-$N$ kernel $K$. Assume that the target function $\tilde{f}$ lies in the $N$-dimensional RKHS corresponding to $K$, then one can obtain an exact formula for the expected value of the test error of the kernel $K$ (but not only the original $K^{(\infty)}$). However, the formula obtained in [2] only takes into account the expectation over the noise $\epsilon$, but not over the samples $x$. Hence, our paper yields a more general result for the test error.

**Upper bound Comparison** To the best of our knowledge, the following result is the closest and most related to ours:

**Theorem 5.1.** *(Proposition 7.4 in [6]) With notation as before, assume, in addition, that: 1) $\tilde{\gamma}_{>M} = 0$, that is, $\tilde{f} \in \mathcal{H}$; 2) for all $k = 1, ..., M$, we have $\sqrt{\sum_{k=1}^M \lambda_k} \leq R$ ; 3) and $N \geq \left(\frac{4}{3} + \frac{R^2}{8\lambda} \log \frac{14R^2}{\lambda\tau}\right)$.*

*Then we have, with a probability of at least $1 - \tau$,*

$$bias \leq 4\lambda \|\tilde{f}\|_{\mathcal{H}}^2;$$
$$variance \leq \frac{8\sigma^2 R^2}{\lambda N} \left(1 + 2\log\frac{2}{\tau}\right).$$

In comparison to [6], Theorem 4.2 makes the following significant improvements:

(i) In Theorem 5.1, as the ridge parameter $\lambda \to 0$, the minimum requirement of $N$ and the upper bound of the variance explode; while in our case, both the minimum requirement of $N$ and the bounds on the variance are independent of $\lambda$.

(ii) While [6] also mentioned the case where $\tilde{\gamma}_{>M} \neq 0$, however their bound for the inconsistent case is implicit; [4] while our work clearly states the impact of $\tilde{\gamma}_{>M}$ on the test error.

(iii) Our upper bounds are sharper than [6]:

(a) For the bias, assume $\tilde{\gamma}_{>M} = 0$ for comparison purposes. Then by line (13) our upper bound would become:

$$\text{bias} \leq \lambda \|\tilde{f}\|_{\mathcal{H}}^2 \left(1 + 2\sqrt{\frac{\log N}{N}}\right) + C_1 \frac{\log N}{N},$$

which improves [6]'s upper bound bias $\leq 4\lambda \|\tilde{f}\|_{\mathcal{H}}^2$ significantly. First, we observe a minor improvement in the constant (by a factor of $\frac{1}{4}$) when $N \to \infty$. Second and more importantly, for finite $N$, we observe a non-asymptotic decay on the upper bound, while [6]'s upper bound is insensitive to $N$.

(b) For the variance, if we replace $\tau$ by $2/N$ in [6]'s upper bound: with probability of at least $1 - 2/N$,

$$\text{variance} \leq \frac{8\sigma^2 R^2}{\lambda N}(1 + 2\log N), \tag{15}$$

which has a decay rate of $\frac{\log N}{N}$, our work shows a much faster decay of $\frac{1}{N}$. Moreover, we show that the "unscaled" variance, that is $N \cdot$ (variance), is actually bounded by $\mathcal{O}\left(1 + \sqrt{\frac{\log N}{N}}\right)$, while [6]'s upper bound on it would explode.

(iv) We also provide a lower bound for the test error that matches with the upper bound when $N \to \infty$, while [6] does not derive any lower bound.

While our work offers overall better bounds, there are some technical differences between the bounds of [6] and ours:

(i) We require some explicit mild condition on input distribution $\rho$ and on kernel $K$, while [6] rely on the assumption $\sqrt{\sum_{k=1}^{M} \lambda_k} \leq R$ which gives an alternative implicit requirement;

(ii) Our work eventually assumes under-parametrization $M < N$, while [6] does not requires this. Essentially, we exploit the assumption on the under-parameterized regime to obtain a sharper bound than [6]. In short, we select the $L_\rho^2$ basis $\psi_k$ for analysis while [6] selects the basis in RKHS, which eventually affects the assumptions and conclusions. For more details, see Appendix C.

**Nyström Subsampling and Random Feature Model**   Nyström Subsampling and Random feature models are two setting closely related to our work. [34, 35] bound the test error from above where the regularizer $\lambda = \lambda(N)$ decays as $N \to \infty$. In comparison, one main contribution of our work is that we provide both tighter upper bound and tighter lower bound than these prior works (they derive the same convergence rate on the upper bound up to constants but they do not derive a lower bound). Another major difference is that our bounds work for any regularization independent to $N$.

---

[4] For inconsistent cases (or misspecified model), [6] only derives an upper bound on the expected value of test error. Also, the bound involves a function defined as an infimum of some set.

**Effective Dimension**  In [34, 35] and other related works on kernel, the quantity $\mathcal{N}(\lambda) = \sum_{k=1}^{M} \frac{\lambda_k}{\lambda_k + \lambda} = \mathrm{Tr}\left[\bar{\mathbf{P}}\right]$ is called the effective dimension and it appeared as a factor in the upper bound. In our work, we can define a related quantity $\mathcal{N}^2(\lambda) = \sum_{k=1}^{M} \frac{\lambda_k^2}{(\lambda_k + \lambda)^2} = \mathrm{Tr}\left[\bar{\mathbf{P}}^2\right]$, which appears in our bound (See Proposition C.14 for details.). Note that $\mathcal{N}^2(\lambda) \leq \mathcal{N}(\lambda) \leq M$. Indeed, we can sharpen the factor $\frac{M}{N}$ in line (10) to $\frac{\mathcal{N}^2(\lambda)}{N}$.

## 5.3  Experiments

We examine the test error and showcase the validity of the bounds we derived for two different finite-rank kernels: the truncated neural tangent kernel (tNTK) and the Legendre kernel (LK) (see details of their definitions in Appendix D). Figure 1 shows the true test error as a function of the sample size $N$. Figure 2 plots our upper bound compared to [6], clearly demonstrating significant improvements. We for instance note that the bound is tighter for any value of $\lambda$. Finally, Figure 9 shows both the upper and lower bound closely enclose the true test error. For more details on the experiment setup, including evaluating the lower bound, please see Appendix D.

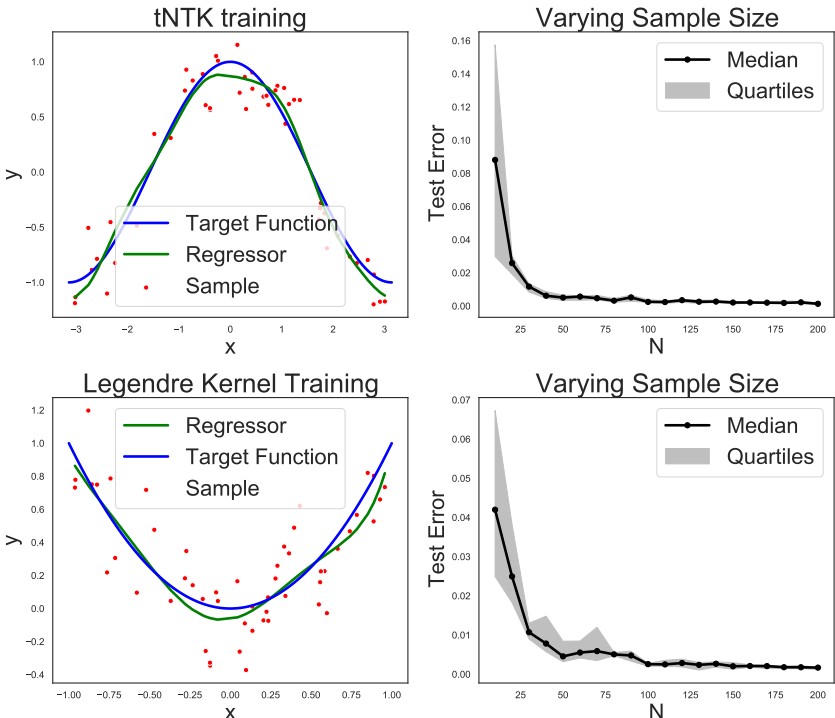

Figure 1: KRR on a finite-rank kernel. Left: KRR training; right: test error for varying $N$, over 10 iterations. The upper and lower quartiles are used as error bars.

## 5.4  Future Research Direction

An interesting extension would be to combine our results with [36, 10] in the over-parameterized regime to give an accurate transition and observe double descent. However, since we are interested in the sharpest possible bounds for the under-parameterized regime, we treat these cases separately. We will do the same for the over-parameterized regime in future work. A special case is where the parameterization is at the threshold $M = N$ and $\lambda = 0$. The blow-up of variance as $M \rightarrow N$ is well-known, where [5, 9] gives empirical and theoretical reports on the blow-up for kernels. We expect that we can formally prove this result for any finite-rank kernels using our exact formula in Proposition C.7 and some anti-concentration inequalities on random matrices.

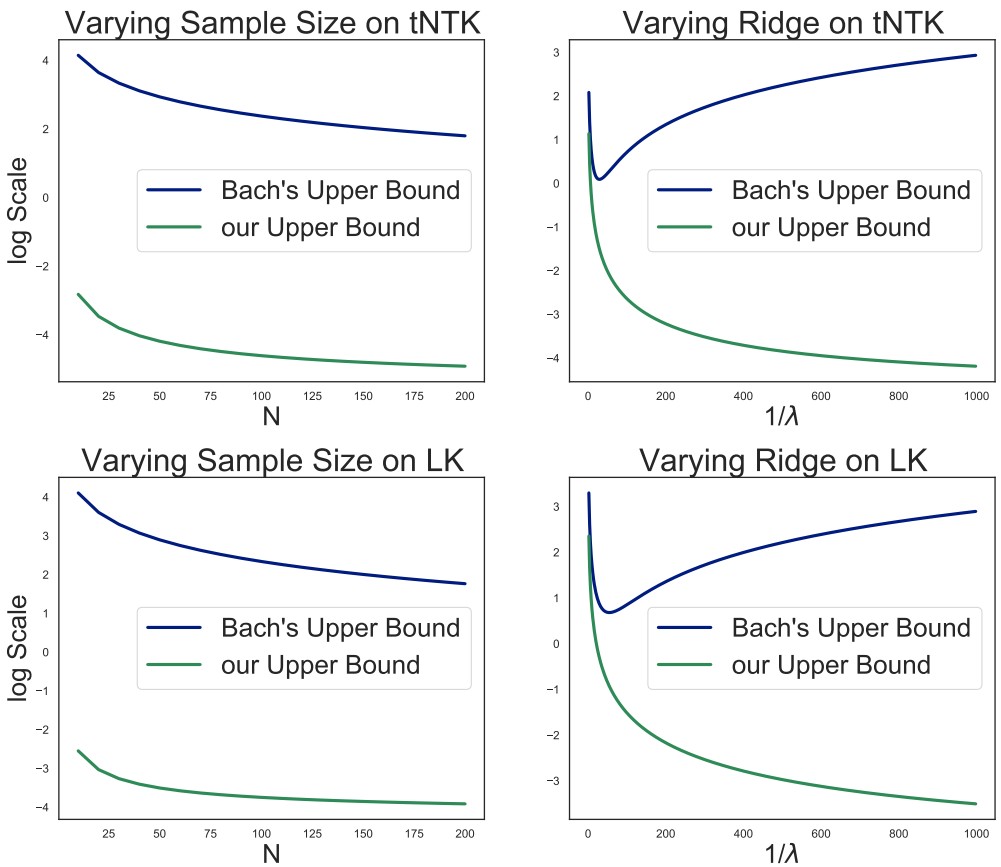

Figure 2: Comparison of test error bounds. Left: upper bounds for varying $N$; right: upper bounds for varying $\lambda$. In natural log scale.

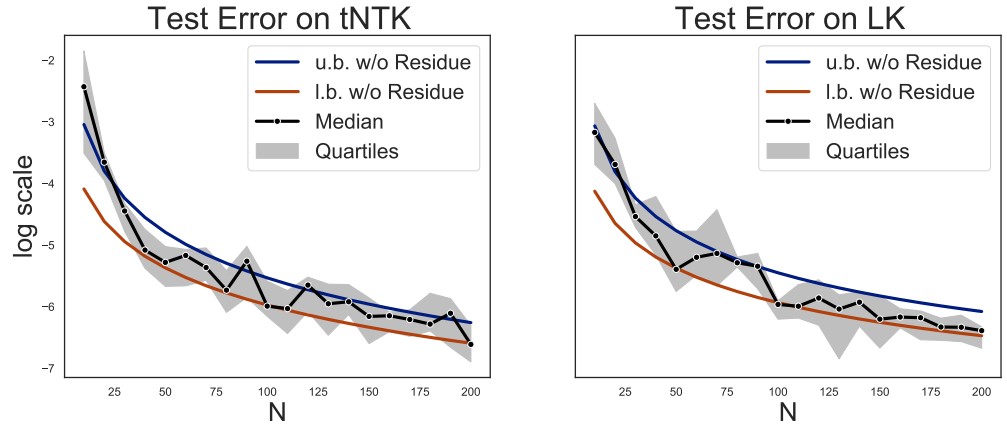

Figure 3: Our bounds (u.b. = upper bound, l.w. = lower bound) comparing to the test error with varying $N$, over 10 iterations, in natural log scale. The residues with coefficients $C_1$ and $C_2$ are dropped by simplicity. As a result, the bounds are not 'bounding' the averaged test error for small $N$. But for large $N$, the residues become negligible.

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

# Appendix

## A Glossary

Table 2: Glossary

| Name | Notation | Expression | Dimension |
|---|---|---|---|
| sampling distribution | $\rho$ | - | $\mathcal{X} \to \mathbb{R}^+$ |
| sampling size | $N$ | - | integer |
| input matrix | $\mathbf{X}$ | $(x_i)_{i=1}^N \underset{iid}{\sim} \rho_{\mathcal{X}}$ | $N \times d$ |
| output vector | $\mathbf{y}$ | $(y_i)_{i=1}^N$ | $N \times 1$ |
| sample | $\mathbf{Z}$ | $(\mathbf{X}, \mathbf{y})$ | $N \times (d+1)$ |
| noise | $\varepsilon$ | - | random scalar |
| noise variance | $\sigma^2$ | $\mathbb{E}[\varepsilon^2]$ | scalar |
| ridge | $\lambda$ | - | scalar |
| finite-rank kernel | $K$ | $\sum_{k=1}^M \lambda_k \psi_k(\cdot)\psi_k(\cdot)$ | $\mathcal{X} \times \mathcal{X} \to \mathbb{R}$ |
| kernel rank | $M$ | - | integer |
| $k$th eigenfunction | $\psi_k$ | - | $\mathcal{X} \to \mathbb{R}$ |
| $k$th value | $\lambda_k$ | - | scalar |
| - | $\boldsymbol{\psi}(x)$ | $[\psi_k(x)]_{k=1}^M$ | $M \times 1$ |
| - | $\boldsymbol{\Psi}$ | $[\psi_k(x_i)]_{k,i}$ | $M \times N$ |
| - | $\boldsymbol{\Lambda}$ | $\operatorname{diag}\left[\lambda_k\right]$ | $M \times M$ |
| kernel matrix | $\mathbf{K}$ | $[K(x_i, x_j)]_{i,j} = \boldsymbol{\Psi}^\top \boldsymbol{\Lambda} \boldsymbol{\Psi}$ | $N \times N$ |
| resolvent | $\mathbf{R}$ | $(\mathbf{K} + \lambda N \mathbf{I}_N)^{-1}$ | $N \times N$ |
| target function | $\tilde{f}$ | $\sum_{k=1}^M \tilde{\gamma}_k \psi_k + \tilde{\gamma}_{>M}\psi_{>M}$ | $\mathcal{X} \to \mathbb{R}$ |
| - | $\tilde{f}_{\leq M}$ | $\sum_{k=1}^M \tilde{\gamma}_k \psi_k$ | $\mathcal{X} \to \mathbb{R}$ |
| $k$th target coefficient | $\tilde{\gamma}_k$ | $\int_{\mathcal{X}} \tilde{f}(x)\psi_k(x)d\rho_{\mathcal{X}}(x)$ | scalar |
| - | $\boldsymbol{\gamma}$ | $[\gamma_k]$ | $M \times 1$ |
| orthonormal complement | $\psi_{>M}$ | - | $\mathcal{X} \to \mathbb{R}$ |
| complementary coefficient | $\tilde{\gamma}_{>M}$ | - | scalar |
| - | $\boldsymbol{\Psi}_{>M}$ | $[\psi_{>M}(x_i)]$ | $1 \times N$ |
| test error | $\mathcal{R}_{\mathbf{Z},\lambda}$ | $\mathbb{E}_{x,\epsilon}\left[(f_{\mathbf{Z},\lambda}(x) - \tilde{f}(x))^2\right]$ | scalar |
| bias | - | $\int_{\mathcal{X}} \left(f_{(\mathbf{X},\tilde{f}(\mathbf{X})),\lambda}(x) - \tilde{f}(x)\right)^2 d\rho(x)$ | scalar |
| variance | - | $\mathcal{R}_{\mathbf{Z},\lambda} - \text{bias} = \mathbb{E}_{x,\varepsilon}\left(\mathbf{K}_x^\top \mathbf{R}\boldsymbol{\varepsilon}\right)^2$ | scalar |
| fluctuation matrix | $\boldsymbol{\Delta}$ | $\frac{1}{N}\boldsymbol{\Psi}\boldsymbol{\Psi}^\top - \mathbf{I}_M$ | $M \times M$ |
| fluctuation | $\delta$ | $\|\boldsymbol{\Delta}\|_{\text{op}}$ | scalar |
| error vector | $\boldsymbol{E}$ | $[\eta_k]$ | $M \times 1$ |
| - | $\eta_k$ | $\frac{1}{N}\sum_{i=1}^N \psi_k(x_i)\psi_{>M}(x_i)$ | scalar |
| - | $\mathbf{B}$ | $(\mathbf{I}_M + \boldsymbol{\Delta} + \lambda\boldsymbol{\Lambda}^{-1})^{-1}$ | $M \times M$ |
| - | $\bar{\mathbf{P}}$ | $\operatorname{diag}\left[\frac{\lambda_k}{\lambda_k + \lambda}\right]$ | $M \times M$ |

# B  Classical KRR Theory

In an effort to keep our manuscript as self-contained as possible, we recall the Mercer decomposition, representer theorem for kernel ridge regression as well as the form of the bias-variance tradeoff in the KRR context.

## B.1  Mercer Decomposition

We begin with a general kernel $K^{(\infty)} : \mathcal{X} \times \mathcal{X} \to \mathbb{R}$.

**Proposition B.1.** *[12] Fix a sample distribution $\rho$. Let $K^{(\infty)} : \mathcal{X} \times \mathcal{X} \to \mathbb{R}$ be a reproducing kernel with corresponding RKHS $\mathcal{H}^{(\infty)}$. There exists a decreasing sequence of real numbers $\lambda_1 \geq \lambda_2 \geq ...$, called the eigenvalues of the kernel $K^{(\infty)}$; and a sequence of pairwise-orthonormal functions $\{\psi_k\}_{k=1}^{\infty} \subset L_\rho^2$, called the eigenfunctions of $K^{(\infty)}$, such that for all $x, x' \in \mathcal{X}$, we have*

$$K^{(\infty)}(x, x') = \sum_{k=1}^{\infty} \lambda_k \psi_k(x) \psi_k(x') \tag{16}$$

In particular, we assume $\lambda_k = 0, \forall k > M$. In this case, we say the kernel $K(x, x') = \sum_{k=1}^{M} \lambda_k \psi_k(x) \psi_k(x')$ is of finite rank $M$ with corresponding (finite-dimensional) RKHS $\mathcal{H}$, recovering equation (2).

The first of these results, allows us to explicitly express the finite-rank kernel ridge regressor $f_{\mathbf{Z},\lambda}$.

**Proposition B.2** (Representer Theorem - [39, Chapter 12]). *Let $\mathbf{R} \overset{\text{def.}}{=} (\mathbf{K} + \lambda N \mathbf{I}_N)^{-1} \in \mathbb{R}^{N \times N}$ be the resolvent matrix and recall the kernel ridge regressor $f_{\mathbf{Z},\lambda}$ given by equation (3):*

$$f_{\mathbf{Z},\lambda} \overset{\text{def.}}{=} \arg\min_{f \in \mathcal{H}} \frac{1}{N} \sum_{i=1}^{N} (f(x_i) - y_i)^2 + \lambda \|f\|_{\mathcal{H}}^2$$

*Then, for every $x \in \mathcal{X}$, we have the expression*

$$f_{\mathbf{Z},\lambda}(x) = \mathbf{y}^\top \mathbf{R} \mathbf{K}_x, \ \forall x \in \mathcal{X}, \tag{17}$$

*where $\mathbf{K}_x \overset{\text{def.}}{=} [K(x_i, x)]_{i=1}^{N} \in \mathbb{R}^{N \times 1}$.*

## B.2  Compact Matrix Expression

First, let $\mathbf{\Psi} \overset{\text{def.}}{=} (\psi_k(x_i))_{k=1,i=1}^{M,N}$ be the random $M \times N$ matrix defined by evaluating the $M$ eigenfunctions on all input training instances $\mathbf{X} \overset{\text{def.}}{=} (x_i)_{i=1}^{N}$, $\mathbf{\Lambda} \overset{\text{def.}}{=} \text{diag}[\lambda_k] \in \mathbb{R}^{M \times M}$, and $\boldsymbol{\psi}(x) \overset{\text{def.}}{=} [\psi_k(x)]_{k=1}^{M} \in \mathbb{R}^{M \times 1}$. The advantage of this notation is that we can rewrite the equations in a more compact form. For equation (17):

$$f_{\mathbf{Z},\lambda}(x) = \mathbf{y}^\top \underbrace{(\mathbf{\Psi}^\top \mathbf{\Lambda} \mathbf{\Psi} + \lambda N \mathbf{I}_M)^{-1}}_{\mathbf{R}} \mathbf{\Psi}^\top \mathbf{\Lambda} \boldsymbol{\psi}(x); \tag{18}$$

for equation (4):

$$\tilde{f}(x) = \tilde{\boldsymbol{\gamma}}^\top \boldsymbol{\psi}(x) + \tilde{\gamma}_{>M} \psi_{>M}(x). \tag{19}$$

Last but not least, we define some important quantities for later analysis.

**Definition B.3** (Fluctuation matrix). *The fluctuation matrix is the random $M \times M$-matrix given by $\mathbf{\Delta} \overset{\text{def.}}{=} \frac{1}{N} \mathbf{\Psi} \mathbf{\Psi}^\top - \mathbf{I}_M$. Our analysis will often involve the operator norm of $\mathbf{\Delta}$, which we denote by $\delta \overset{\text{def.}}{=} \|\mathbf{\Delta}\|_{op}$.*

The fluctuation matrix $\mathbf{\Delta}$ measures the first source of randomness in the KRR's test error. Namely it encodes the degree of non-orthonormality between the vectors obtained by evaluating of the $M$ eigenfunctions $\psi_1, \ldots, \psi_M$ on the input $\mathbf{X}$.

The second source of randomness in the KRR's test error comes from the empirical evaluation of the dot product of the eigenfunction $\psi_k$'s and the orthogonal complement $\psi_{>M}$:

**Definition B.4** (Error Vector). *$\boldsymbol{E} \overset{\text{def.}}{=} \frac{1}{N} \mathbf{\Psi} \psi_{>M}(\mathbf{X})$ is called the error vector.*

The random matrix $\mathbf{\Delta}$ and the random vector $\boldsymbol{E}$ are centered; i.e. $\mathbb{E}_{\mathbf{X}}[\mathbf{\Delta}] = 0$ and $\mathbb{E}_{\mathbf{X}}[\boldsymbol{E}] = 0$.

### B.3 Bias-Variance Decomposition

The derivation of several contemporary KRR generalization bounds [6, 26, 27] involves the classical Bias-Variance Trade-off:

**Proposition B.5** (Bias-Variance Trade-off)**.** *Fix a sample* $\mathbf{Z}$*. Recall the definition 3.4 of test error* $\mathcal{R}_{\mathbf{Z},\lambda}$*, bias, and variance:*

$$\mathcal{R}_{\mathbf{Z},\lambda} \overset{\text{def.}}{=} \mathbb{E}_{x,\epsilon}\left[(f_{\mathbf{Z},\lambda}(x) - \tilde{f}(x))^2\right] = \mathbb{E}_{\epsilon}\left[\int_{\mathcal{X}}\left(f_{\mathbf{Z},\lambda}(x) - \tilde{f}(x)\right)^2 d\rho(x)\right];$$

$$bias \overset{\text{def.}}{=} \int_{\mathcal{X}}\left(f_{(\mathbf{X},\tilde{f}(\mathbf{X})),\lambda}(x) - \tilde{f}(x)\right)^2 d\rho(x);$$

$$variance \overset{\text{def.}}{=} \mathcal{R}_{\mathbf{Z},\lambda} - bias.$$

*Then, we can write* $variance_{test} = \mathbb{E}_{x,\varepsilon}\left(\mathbf{K}_x^\top \mathbf{R}\boldsymbol{\varepsilon}\right)^2$ *and hence the test error* $\mathcal{R}_{\mathbf{Z},\lambda}$ *admits a decomposition:*

$$R_{\mathbf{Z},\lambda} = bias + \mathbb{E}_{x,\varepsilon}\left(\mathbf{K}_x^\top \mathbf{R}\boldsymbol{\varepsilon}\right)^2.$$

*Proof.* See the proof of Theorem C.8. $\qquad\square$

## C  Proofs

In this section, we will derive the essential lemmata and propositions for proving the main theorems.

### C.1  Formula Derivation

We begin with writing the test error in convenient forms.

#### C.1.1  Bias

We first derive, from the definition of the bias, a convenient expression to proceed:

**Proposition C.1** (Bias Expression)**.** *Let* $\boldsymbol{\Psi}_{>M} \overset{\text{def.}}{=} [\psi_{>M}(x_i)]_{i=1}^N$ *as an* $1 \times N$*- row vector,* $\left(\begin{smallmatrix}\boldsymbol{\Psi} \\ \boldsymbol{\Psi}_{>M}\end{smallmatrix}\right)$ *as an* $(M+1) \times N$ *matrix. Denote* $\mathbf{P} \overset{\text{def.}}{=} \left(\mathbf{P}_{\leq M} \quad \mathbf{P}_{>M}\right) = \boldsymbol{\Lambda}\boldsymbol{\Psi}\mathbf{R}\left(\boldsymbol{\Psi}^\top \quad \boldsymbol{\Psi}_{>M}^\top\right) \in \mathbb{R}^{M \times (M+1)}$, $\mathbf{P}_{\leq M} \in \mathbb{R}^{M \times M}$ *and* $\mathbf{P}_{>M} \in \mathbb{R}^{M \times 1}$*. Then the bias admits the following expression:*

$$bias = \underbrace{\tilde{\gamma}_{>M}^2}_{\textit{Finite Rank Error}} + \underbrace{\|\tilde{\boldsymbol{\gamma}} - \mathbf{P}_{\leq M}\tilde{\boldsymbol{\gamma}} - \tilde{\gamma}_{>M}\mathbf{P}_{>M}\|_2^2}_{\textit{Fitting Error}}.$$

*Proof.* Recall that, by equations (18) and (19), we can write

$$\tilde{f}(x) = \tilde{\boldsymbol{\gamma}}^\top \boldsymbol{\psi}(x) + \tilde{\gamma}_{>M}\psi_{>M}(x),$$

$$f_{(\mathbf{X},\tilde{f}(\mathbf{X})),\lambda}(x) = (\tilde{\boldsymbol{\gamma}}^\top \boldsymbol{\Psi} + \tilde{\gamma}_{>M}\boldsymbol{\Psi}_{>M}^\top)\mathbf{R}\boldsymbol{\Psi}^\top \boldsymbol{\Lambda}\boldsymbol{\Psi}(x).$$

Hence

$$\begin{aligned}
bias &= \mathbb{E}_x\left[\left(\tilde{\boldsymbol{\gamma}}^\top \boldsymbol{\psi}(x) + \tilde{\gamma}_{>M}\psi_{>M}(x) - (\tilde{\boldsymbol{\gamma}}^\top \boldsymbol{\Psi} + \tilde{\gamma}_{>M}\boldsymbol{\Psi}_{>M}^\top)\mathbf{R}\boldsymbol{\Psi}^\top \boldsymbol{\Lambda}\boldsymbol{\Psi}(x)\right)^2\right] \\
&= \left\|\begin{pmatrix}\tilde{\boldsymbol{\gamma}} \\ \tilde{\gamma}_{>M}\end{pmatrix} - \begin{pmatrix}\mathbf{P}\begin{pmatrix}\tilde{\boldsymbol{\gamma}} \\ \tilde{\gamma}_{>M}\end{pmatrix} \\ 0\end{pmatrix}\right\|_2^2 \qquad\qquad (20) \\
&= \underbrace{\tilde{\gamma}_{>M}^2}_{\textit{Finite Rank Error}} + \underbrace{\|\tilde{\boldsymbol{\gamma}} - \mathbf{P}_{\leq M}\tilde{\boldsymbol{\gamma}} - \tilde{\gamma}_{>M}\mathbf{P}_{>M}\|_2^2}_{\textit{Fitting Error}},
\end{aligned}$$

in line (20), we use Parseval's identity. $\qquad\square$

We proceed by reformulating the projection matrix $\mathbf{P}$, first with the left matrix $\mathbf{P}_{\leq M}$:

**Lemma C.2.** *Recall the following notations*

$$\mathbf{K} \stackrel{\text{def.}}{=} \boldsymbol{\Psi}^\top \boldsymbol{\Lambda} \boldsymbol{\Psi}, \quad \mathbf{R} \stackrel{\text{def.}}{=} (\mathbf{K} + \lambda N \mathbf{I}_M)^{-1}, \quad \mathbf{P}_{\leq M} \stackrel{\text{def.}}{=} \boldsymbol{\Lambda} \boldsymbol{\Psi} \mathbf{R} \boldsymbol{\Psi}^\top.$$

*Define the symmetric random matrix* $\mathbf{B} \stackrel{\text{def.}}{=} (\mathbf{I}_M + \boldsymbol{\Delta} + \lambda \boldsymbol{\Lambda}^{-1})^{-1}$. *It holds that*

$$\mathbf{P}_{\leq M} = \mathbf{I}_M - \lambda \mathbf{B} \boldsymbol{\Lambda}^{-1}.$$

*Proof.* We first observe that

$$\boldsymbol{\Psi} \boldsymbol{\Psi}^\top \mathbf{P}_{\leq M} = \boldsymbol{\Psi} \boldsymbol{\Psi}^\top \boldsymbol{\Lambda} \boldsymbol{\Psi} \mathbf{R} \boldsymbol{\Psi}^\top \tag{21}$$

$$= \boldsymbol{\Psi} \mathbf{K} (\mathbf{K} + \lambda N \mathbf{I}_M)^{-1} \boldsymbol{\Psi}^\top$$

$$= \boldsymbol{\Psi} \left( \mathbf{I}_M - \lambda N (\mathbf{K} + \lambda N \mathbf{I}_M)^{-1} \right) \boldsymbol{\Psi}^\top$$

$$= \boldsymbol{\Psi} \boldsymbol{\Psi}^\top - \lambda N \boldsymbol{\Psi} (\mathbf{K} + \lambda N \mathbf{I}_M)^{-1} \boldsymbol{\Psi}^\top. \tag{22}$$

From lines (21)-(22) and the definition of the fluctuation matrix $\boldsymbol{\Delta}$ we deduce

$$\frac{1}{N} \boldsymbol{\Psi} \boldsymbol{\Psi}^\top (\mathbf{I}_M - \mathbf{P}_{\leq M}) = \lambda \boldsymbol{\Psi} (\mathbf{K} + \lambda N \mathbf{I}_M)^{-1} \boldsymbol{\Psi}^\top$$

$$(\mathbf{I}_M + \boldsymbol{\Delta})(\mathbf{I}_M - \mathbf{P}_{\leq M}) = \lambda \boldsymbol{\Psi} \mathbf{R} \boldsymbol{\Psi}^\top$$

$$(\mathbf{I}_M + \boldsymbol{\Delta})(\mathbf{I}_M - \mathbf{P}_{\leq M}) = \lambda \boldsymbol{\Lambda}^{-1} \mathbf{P}_{\leq M}$$

$$(\boldsymbol{\Lambda} + \boldsymbol{\Lambda} \boldsymbol{\Delta})(\mathbf{I}_M - \mathbf{P}_{\leq M}) = \lambda \mathbf{P}_{\leq M}$$

$$\boldsymbol{\Lambda} + \boldsymbol{\Lambda} \boldsymbol{\Delta} = (\boldsymbol{\Lambda} + \boldsymbol{\Lambda} \boldsymbol{\Delta} + \lambda \mathbf{I}_M) \mathbf{P}_{\leq M}. \tag{23}$$

Rearranging (23) and applying the definition of $B$ we find that

$$\mathbf{P}_{\leq M} = (\boldsymbol{\Lambda} + \boldsymbol{\Lambda} \boldsymbol{\Delta} + \lambda \mathbf{I}_M)^{-1}(\boldsymbol{\Lambda} + \boldsymbol{\Lambda} \boldsymbol{\Delta}) \tag{24}$$

$$= \mathbf{I}_M - \lambda (\boldsymbol{\Lambda} + \boldsymbol{\Lambda} \boldsymbol{\Delta} + \lambda \mathbf{I}_M)^{-1}$$

$$= \mathbf{I}_M - \lambda (\boldsymbol{\Lambda} + \boldsymbol{\Lambda} \boldsymbol{\Delta} + \lambda \mathbf{I}_M)^{-1} \boldsymbol{\Lambda} \boldsymbol{\Lambda}^{-1}$$

$$= \mathbf{I}_M - \lambda \mathbf{B} \boldsymbol{\Lambda}^{-1}. \tag{25}$$

$\square$

Arguing analogously for the right matrix $\mathbf{P}_{>M}$, we draw the subsequent similar conclusion.

**Lemma C.3.** *Recall the following notations*

$$\mathbf{K} \stackrel{\text{def.}}{=} \boldsymbol{\Psi}^\top \boldsymbol{\Lambda} \boldsymbol{\Psi}, \quad \mathbf{R} \stackrel{\text{def.}}{=} (\mathbf{K} + \lambda N \mathbf{I}_M)^{-1}, \quad \mathbf{P}_{>M} \stackrel{\text{def.}}{=} \boldsymbol{\Lambda} \boldsymbol{\Psi} \mathbf{R} \boldsymbol{\Psi}_{>M}^\top,$$

$$E \stackrel{\text{def.}}{=} \frac{1}{N} \boldsymbol{\Psi} \boldsymbol{\Psi}_{>M}^\top, \quad \mathbf{B} \stackrel{\text{def.}}{=} (\mathbf{I}_M + \boldsymbol{\Delta} + \lambda \boldsymbol{\Lambda}^{-1})^{-1}.$$

*We have that* $\mathbf{P}_{>M} = \mathbf{B} E$.

*Proof.* Similarly to (21)-(22) we note that

$$\boldsymbol{\Psi} \boldsymbol{\Psi}^\top \mathbf{P}_{>M} = \boldsymbol{\Psi} \boldsymbol{\Psi}^\top \boldsymbol{\Lambda} \boldsymbol{\Psi} \mathbf{R} \boldsymbol{\Psi}_{>M}^\top$$

$$= \boldsymbol{\Psi} \mathbf{K} (\mathbf{K} + \lambda N \mathbf{I}_M)^{-1} \boldsymbol{\Psi}_{>M}^\top$$

$$= \boldsymbol{\Psi} \left( \mathbf{I}_M - \lambda N (\mathbf{K} + \lambda N \mathbf{I}_M)^{-1} \right) \boldsymbol{\Psi}_{>M}^\top$$

$$= \boldsymbol{\Psi} \boldsymbol{\Psi}_{>M}^\top - \lambda N \boldsymbol{\Psi} (\mathbf{K} + \lambda N \mathbf{I}_M)^{-1} \boldsymbol{\Psi}_{>M}^\top.$$

Analogously to the computations in (24)-(25)

$$(\mathbf{I}_M + \boldsymbol{\Delta}) \mathbf{P}_{>M} = E - \lambda \boldsymbol{\Psi} (\mathbf{K} + \lambda N \mathbf{I}_M)^{-1} \boldsymbol{\Psi}_{>M}^\top$$

$$(\mathbf{I}_M + \boldsymbol{\Delta}) \mathbf{P}_{>M} = E - \lambda \boldsymbol{\Lambda}^{-1} \boldsymbol{\Lambda} \boldsymbol{\Psi} (\mathbf{K} + \lambda N \mathbf{I}_M)^{-1} \boldsymbol{\Psi}_{>M}^\top$$

$$(\mathbf{I}_M + \boldsymbol{\Delta}) \mathbf{P}_{>M} = E - \lambda \boldsymbol{\Lambda}^{-1} \mathbf{P}_{>M}$$

$$(\boldsymbol{\Lambda} + \boldsymbol{\Lambda} \boldsymbol{\Delta}) \mathbf{P}_{>M} = \boldsymbol{\Lambda} E - \lambda \mathbf{P}_{>M}$$

$$(\boldsymbol{\Lambda} + \boldsymbol{\Lambda} \boldsymbol{\Delta} + \lambda \mathbf{I}_M) \mathbf{P}_{>M} = \boldsymbol{\Lambda} E$$

$$\mathbf{P}_{>M} = (\boldsymbol{\Lambda} + \boldsymbol{\Lambda} \boldsymbol{\Delta} + \lambda \mathbf{I}_M)^{-1} \boldsymbol{\Lambda} E$$

$$\mathbf{P}_{>M} = \mathbf{B} E.$$

$\square$

**Lemma C.4** (Fitting Error). *Recall the notation*

$$fitting\ error = \|\tilde{\gamma} - \mathbf{P}_{\leq M}\tilde{\gamma} - \tilde{\gamma}_{>M}\mathbf{P}_{>M}\|_2^2,$$

$$\mathbf{B} \stackrel{\text{def.}}{=} (\mathbf{I}_M + \mathbf{\Delta} + \lambda\mathbf{\Lambda}^{-1})^{-1}.$$

*We have fitting error* $= \left\|\mathbf{B}\left(\lambda\mathbf{\Lambda}^{-1}\tilde{\gamma} - \boldsymbol{E}\tilde{\gamma}_{>M}\right)\right\|_2^2.$

*Proof.* By lemmata C.2 and C.3,

$$\|\tilde{\gamma} - \mathbf{P}_{\leq M}\tilde{\gamma} - \tilde{\gamma}_{>M}\mathbf{P}_{>M}\|_2^2 = \left\|\tilde{\gamma} - (\mathbf{I}_M - \lambda\mathbf{B}\mathbf{\Lambda}^{-1})\tilde{\gamma}\tilde{\gamma} - \mathbf{B}\boldsymbol{E}\tilde{\gamma}_{>M}\right\|_2^2$$
$$= \left\|\mathbf{B}\left(\lambda\mathbf{\Lambda}^{-1}\tilde{\gamma} - \boldsymbol{E}\tilde{\gamma}_{>M}\right)\right\|_2^2.$$

$\square$

Hence we come up with a new expression of the bias:

**Proposition C.5** (Bias). *Recall that* $\mathbf{B} \stackrel{\text{def.}}{=} (\mathbf{I}_M + \mathbf{\Delta} + \lambda\mathbf{\Lambda}^{-1})^{-1}$. *The bias* $\mathbb{E}_x\left(f_{\mathbf{X}}^{\lambda}(x) - \tilde{f}(x)\right)^2$ *has the following expression:*

$$bias = \tilde{\gamma}_{>M}^2 + \left\|\mathbf{B}\left(\lambda\mathbf{\Lambda}^{-1}\tilde{\gamma} - \tilde{\gamma}_{>M}\boldsymbol{E}\right)\right\|_2^2.$$

*Proof.* We apply Proposition C.1 and Lemma C.4 to obtain the result. $\square$

### C.1.2 Variance

If we consider noise in the label, we have to compute the variance part of the test error.

**Proposition C.6** (Variance Expression). *Define*

$$\mathbf{M} \stackrel{\text{def.}}{=} \mathbb{E}_x[\mathbf{K}_x\mathbf{K}_x^{\top}]$$
$$= \mathbb{E}_x[\mathbf{\Psi}^{\top}\mathbf{\Lambda}\boldsymbol{\psi}(x)\boldsymbol{\psi}(x)^{\top}\mathbf{\Lambda}\mathbf{\Psi}]$$
$$= \mathbf{\Psi}^{\top}\mathbf{\Lambda}\mathbb{E}_x[\boldsymbol{\psi}(x)\boldsymbol{\psi}(x)^{\top}]\mathbf{\Lambda}\mathbf{\Psi}$$
$$= \mathbf{\Psi}^{\top}\mathbf{\Lambda}\mathbf{I}_M\mathbf{\Lambda}\mathbf{\Psi}$$
$$= \mathbf{\Psi}^{\top}\mathbf{\Lambda}^2\mathbf{\Psi}.$$

*We can further simplify the variance part:*

$$variance \stackrel{\text{def.}}{=} \mathbb{E}_{x,\varepsilon}\left[\left(\mathbf{K}_x^{\top}\mathbf{R}\varepsilon\right)^2\right]$$
$$= \mathbb{E}_{x,\varepsilon}\left[\varepsilon^{\top}\mathbf{R}\mathbf{K}_x\mathbf{K}_x^{\top}\mathbf{R}\varepsilon\right]$$
$$= \mathbb{E}_{\varepsilon}\left[\varepsilon^{\top}\mathbf{R}\mathbf{M}\mathbf{R}\varepsilon\right]$$
$$= \sigma^2 \cdot \text{Tr}[\mathbf{R}\mathbf{M}\mathbf{R}].$$

**Theorem C.7** (Variance). *Recall that* $\mathbf{B} \stackrel{\text{def.}}{=} (\mathbf{I}_M + \mathbf{\Delta} + \lambda\mathbf{\Lambda}^{-1})^{-1}$. *The variance part, variance, can be expressed as:*

$$variance = \frac{\sigma^2}{N} \text{Tr}\left[\mathbf{B}^2(\mathbf{I}_M + \mathbf{\Delta})\right].$$

*Proof.* We argue similarly as in lemma C.2. Since

$$\mathbf{\Psi}\mathbf{\Psi}^{\top}\mathbf{\Lambda}\mathbf{\Psi}\mathbf{R} = \mathbf{\Psi}\mathbf{K}(\mathbf{K} + \lambda N\mathbf{I}_M)^{-1}$$
$$= \mathbf{\Psi}(\mathbf{I}_M - \lambda N\mathbf{R})$$
$$= \mathbf{\Psi} - \lambda N\mathbf{\Psi}\mathbf{R},$$

therefore, we deduce that

$$(\mathbf{I}_M + \boldsymbol{\Delta})\boldsymbol{\Lambda}\boldsymbol{\Psi}\mathbf{R} = \frac{1}{N}\boldsymbol{\Psi} - \lambda\boldsymbol{\Psi}\mathbf{R} \tag{26}$$

$$(\mathbf{I}_M + \boldsymbol{\Delta})\boldsymbol{\Lambda}\boldsymbol{\Psi}\mathbf{R} = \frac{1}{N}\boldsymbol{\Psi} - \lambda\boldsymbol{\Lambda}^{-1}\boldsymbol{\Lambda}\boldsymbol{\Psi}\mathbf{R}$$

$$(\mathbf{I}_M + \boldsymbol{\Delta} + \lambda\boldsymbol{\Lambda}^{-1})\boldsymbol{\Lambda}\boldsymbol{\Psi}\mathbf{R} = \frac{1}{N}\boldsymbol{\Psi}$$

$$\boldsymbol{\Lambda}\boldsymbol{\Psi}\mathbf{R} = \frac{1}{N}(\mathbf{I}_M + \boldsymbol{\Delta} + \lambda\boldsymbol{\Lambda}^{-1})^{-1}\boldsymbol{\Psi}$$

$$\boldsymbol{\Lambda}\boldsymbol{\Psi}\mathbf{R} = \frac{1}{N}\mathbf{B}\boldsymbol{\Psi}. \tag{27}$$

By leveraging the identity $\mathbf{M} = \boldsymbol{\Psi}^\top\boldsymbol{\Lambda}^2\boldsymbol{\Psi}$ and elementary properties of the trace map, the computations in (26)-(27) imply that

$$\mathrm{Tr}[\mathbf{R}\mathbf{M}\mathbf{R}] = \mathrm{Tr}[\mathbf{R}\boldsymbol{\Psi}^\top\boldsymbol{\Lambda}^2\boldsymbol{\Psi}\mathbf{R}] \tag{28}$$

$$= \mathrm{Tr}\left[(\boldsymbol{\Lambda}\boldsymbol{\Psi}\mathbf{R})^\top (\boldsymbol{\Lambda}\boldsymbol{\Psi}\mathbf{R})\right] \tag{29}$$

$$= \mathrm{Tr}\left[\left(\frac{1}{N}\mathbf{B}\boldsymbol{\Psi}\right)^\top \left(\frac{1}{N}\mathbf{B}\boldsymbol{\Psi}\right)\right] \tag{30}$$

$$= \frac{1}{N}\mathrm{Tr}\left[\frac{1}{N}\boldsymbol{\Psi}^\top\mathbf{B}^\top\mathbf{B}\boldsymbol{\Psi}\right]$$

$$= \frac{1}{N}\mathrm{Tr}\left[\mathbf{B}^\top\mathbf{B}\cdot\frac{1}{N}\boldsymbol{\Psi}\boldsymbol{\Psi}^\top\right] \tag{31}$$

$$= \frac{1}{N}\mathrm{Tr}\left[\mathbf{B}^\top\mathbf{B}(\mathbf{I}_M + \boldsymbol{\Delta})\right] \tag{32}$$

$$= \frac{1}{N}\mathrm{Tr}\left[\mathbf{B}^2(\mathbf{I}_M + \boldsymbol{\Delta})\right]; \tag{33}$$

in more detail: in line (28), we use the definition of $\mathbf{M}$; in line (29), we use the fact that both $\boldsymbol{\Lambda}$ and $\mathbf{R}$ are symmetric; in line (30), we use line (27); in line (31), we use the cyclicity of the trace; in line (32), we use the definition of $\boldsymbol{\Delta}$; in line (33), we use the symmetry of $\mathbf{B}$. We obtain the result upon applying Lemma C.6. $\qquad\square$

### C.1.3   Test Error

The Bias-Variance trade-off (see Proposition B.5) decomposed the KRR's test error into two terms, the bias and variance. Since Propositions C.5 and C.7 give us exact expressions for the bias and variance, respectively, we deduce the following exact expression for the KRR's test error.

**Theorem C.8** (Exact Formula for KRR's Test Error). *The test error $\mathcal{R}_{\mathbf{Z},\lambda}$ of KRR equals*

$$\mathcal{R}_{\mathbf{Z},\lambda} = \underbrace{\overbrace{\left\|\mathbf{B}\left(\lambda\boldsymbol{\Lambda}^{-1}\tilde{\boldsymbol{\gamma}} - \tilde{\gamma}_{>M}\boldsymbol{E}_M\right)\right\|_2^2}^{\text{fitting error}} + \overbrace{\tilde{\gamma}_{>M}^2}^{\text{finite rank error}}}_{\text{bias}} + \underbrace{\frac{\sigma_{noise}^2}{N}\mathrm{Tr}\left[\mathbf{B}^2(\mathbf{I}_M + \boldsymbol{\Delta})\right]}_{\text{variance}},$$

*where* $\mathbf{B} \stackrel{\text{def.}}{=} (\mathbf{I}_M + \boldsymbol{\Delta} + \lambda\boldsymbol{\Lambda}^{-1})^{-1}$.

*Proof.* We begin with the bias/variance decomposition:

$$R_{\mathbf{Z}}^\lambda \stackrel{\text{def.}}{=} \mathbb{E}_y\|f_{\mathbf{Z}}^\lambda - \tilde{f}\|_{L_{\rho_\mathcal{X}}^2}^2$$

$$= \mathbb{E}_{x,y}\left(\mathbf{K}_x^\top\mathbf{R}\mathbf{y} - \tilde{f}(x)\right)^2$$

$$= \mathbb{E}_{\varepsilon,x}\left(\mathbf{K}_x^\top\mathbf{R}(\tilde{f}(\mathbf{X}) + \varepsilon) - \tilde{f}(x)\right)^2$$

$$= \mathbb{E}_x\left(f_{\mathbf{X}}^\lambda(x) - \tilde{f}(x)\right)^2 + \mathbb{E}_{x,\varepsilon}\left[\left(\mathbf{K}_x^\top\mathbf{R}\varepsilon\right)^2\right]$$

$$= \text{bias} + \text{variance},$$

then we apply Propositions C.5 and C.7. □

For the validation of the Theorem C.8, please see Appendix D for details.

The matrix $\mathbf{B}$ plays an important role in the expression since it encodes most information of the KRR. Therefore, the following subsection will discuss the approximation of the matrix $\mathbf{B}$.

## C.2 Matrix Approximation

Recall that the matrix $\mathbf{B} \overset{\text{def.}}{=} (\mathbf{I}_M + \boldsymbol{\Delta} + \lambda\boldsymbol{\Lambda}^{-1})^{-1}$ is the inverse of a random matrix. The following lemma helps to approximate $\mathbf{B}$. Informally, it says that: given that $\delta \overset{\text{def.}}{=} \|\boldsymbol{\Delta}\|_{\text{op}} < 1$. We have

$$\mathbf{B} = \sum_{s=0}^{\infty}(-\bar{\mathbf{P}}\boldsymbol{\Delta})^s\bar{\mathbf{P}}$$

in operator norm $\|\cdot\|_{op}$ for an $M \times M$ matrix $\bar{P}$ depending only on the $M$ eigenvalues $\{\lambda_k\}_{k=1}^M$ and on the ridge $\lambda > 0$. More precisely we have the following.

**Lemma C.9** (B-Expansion). *Given that $\delta \overset{\text{def.}}{=} \|\boldsymbol{\Delta}\|_{op} < 1$. It holds that*

$$\lim_{n\uparrow\infty} \left\| \mathbf{B} - \sum_{s=0}^{n}(-\bar{\mathbf{P}}\boldsymbol{\Delta})^s\bar{\mathbf{P}} \right\|_{op} = 0$$

*where $\bar{\mathbf{P}} \overset{\text{def.}}{=} \text{diag}\left[\frac{\lambda_k}{\lambda_k+\lambda}\right]_k = \boldsymbol{\Lambda}(\boldsymbol{\Lambda} + \lambda\mathbf{I}_M)^{-1} \in \mathbb{R}^{M\times M}$.*

*Proof.* Set $\mathbf{A} = \mathbf{I}_M + \lambda\boldsymbol{\Lambda}^{-1}$ and repeatedly use the formula $(\mathbf{A}+\boldsymbol{\Delta})^{-1} = \mathbf{A}^{-1} - \mathbf{A}^{-1}\boldsymbol{\Delta}(\mathbf{A}+\boldsymbol{\Delta})^{-1}$ from [31], we have

$$\begin{aligned}
\mathbf{B} &\overset{\text{def.}}{=} (\mathbf{I}_M + \boldsymbol{\Delta} + \lambda\boldsymbol{\Lambda}^{-1})^{-1} \\
&= (\mathbf{A} + \boldsymbol{\Delta})^{-1} \\
&= \mathbf{A}^{-1} - \mathbf{A}^{-1}\boldsymbol{\Delta}(\mathbf{A} + \boldsymbol{\Delta})^{-1} \\
&= \mathbf{A}^{-1} - \mathbf{A}^{-1}\boldsymbol{\Delta}\left(\mathbf{A}^{-1} - \mathbf{A}^{-1}\boldsymbol{\Delta}(\mathbf{A} + \boldsymbol{\Delta})^{-1}\right) \\
&= \mathbf{A}^{-1} - \mathbf{A}^{-1}\boldsymbol{\Delta}\mathbf{A}^{-1} + (\mathbf{A}^{-1}\boldsymbol{\Delta})^2(\mathbf{A} + \boldsymbol{\Delta})^{-1} \\
&= \sum_{s=0}^{n}(-\mathbf{A}^{-1}\boldsymbol{\Delta})^s\mathbf{A}^{-1} + (-\mathbf{A}^{-1}\boldsymbol{\Delta})^{n+1}(\mathbf{A} + \boldsymbol{\Delta})^{-1}
\end{aligned}$$

Note that $\mathbf{A}^{-1} = (\mathbf{I}_M + \lambda\boldsymbol{\Lambda}^{-1})^{-1} = \boldsymbol{\Lambda}(\boldsymbol{\Lambda} + \lambda\mathbf{I}_M)^{-1} = \bar{\mathbf{P}}$ with operator norm $\frac{\lambda_1}{\lambda_1+\lambda} < 1$, hence we have $(\mathbf{A}^{-1}\boldsymbol{\Delta})^{n+1} = (-\bar{\mathbf{P}}\boldsymbol{\Delta})^{n+1} \to 0$ in operator norm as $n \to \infty$. Hence

$$\begin{aligned}
\mathbf{B} &= \sum_{s=0}^{\infty}(-\mathbf{A}^{-1}\boldsymbol{\Delta})^s\mathbf{A}^{-1} \\
&= \sum_{s=0}^{\infty}(-\bar{\mathbf{P}}\boldsymbol{\Delta})^s\bar{\mathbf{P}}
\end{aligned}$$

in operator norm. □

Due to the convergence result in lemma C.9, it is natural to define:

**Definition C.10.** *For any $n \in \mathbb{N} \cup \{\infty\}$, write $\mathbf{B}^{(n)} = \sum_{s=0}^{n}(-\bar{\mathbf{P}}\boldsymbol{\Delta})^s\bar{\mathbf{P}}$. For example, We have*

$$\begin{aligned}
\mathbf{B}^{(0)} &= \bar{\mathbf{P}} \\
\mathbf{B}^{(1)} &= \bar{\mathbf{P}} - \bar{\mathbf{P}}\boldsymbol{\Delta}\bar{\mathbf{P}} \\
\mathbf{B}^{(\infty)} &= \mathbf{B}
\end{aligned}$$

Although lemma C.9 is valid when $\delta < 1$, we need a slightly stronger condition that $\delta$ is upper bounded by an arbitrary constant strictly small than 1. For simplicity, we assume this constant to be $\frac{1}{2}$ in the following lemma:

**Lemma C.11** (B-Approximation). *Assume that* $\delta \overset{\text{def.}}{=} \|\mathbf{\Delta}\|_{op} < \frac{1}{2}$. *Let* $\mathbf{B}^{(n)} = \sum_{s=0}^{n}(-\bar{\mathbf{P}}\mathbf{\Delta})^s\bar{\mathbf{P}}$ *be the $n$th-order approximation of the matrix $\mathbf{B}$ as in definition C.10. Then we have*

$$\left\|\mathbf{B} - \mathbf{B}^{(n)}\right\|_{op} < 2\delta^{n+1}.$$

*Proof.* We first bound the operator norm of the matrix $\mathbf{B}$: since the minimum singular value of the matrix $\bar{\mathbf{P}}^{-1} + \mathbf{\Delta}$ is at least

$$\frac{\lambda_k + \lambda}{\lambda_k} - \|\Delta\|_{op} \geq 1 + \frac{\lambda}{\lambda_k} - \frac{1}{2} > \frac{1}{2},$$

and hence

$$\|\mathbf{B}\|_{op} = \left\|\left(\bar{\mathbf{P}}^{-1} + \mathbf{\Delta}\right)^{-1}\right\|_{op} < 2.$$

Also, we have

$$\mathbf{B} - \mathbf{B}^{(n)} = \sum_{s=n+1}^{\infty}(-\bar{\mathbf{P}}\mathbf{\Delta})^s\bar{\mathbf{P}}$$

$$= (-\bar{\mathbf{P}}\mathbf{\Delta})^{n+1}\sum_{s=0}^{\infty}(-\bar{\mathbf{P}}\mathbf{\Delta})^s\bar{\mathbf{P}}$$

$$= (-\bar{\mathbf{P}}\mathbf{\Delta})^{n+1}\mathbf{B}.$$

Hence $\left\|\mathbf{B} - \mathbf{B}^{(n)}\right\|_{op} \leq \left\|\bar{\mathbf{P}}\mathbf{\Delta}\right\|_{op}^{n+1}\|\mathbf{B}\|_{op} < \|\mathbf{\Delta}\|_{op}^{n+1} \cdot 2 = 2\delta^{n+1}$, since we have $\left\|\bar{\mathbf{P}}\right\|_{op} = \frac{\lambda_1}{\lambda_1 + \lambda} < 1$. $\qquad\square$

Note that the upper bound $\frac{1}{2}$ of $\delta$ can be replaced by any constant strictly small than 1 to get a similar conclusion.

**Remark C.12.** *Using the concentration result from random matrix theory, for $M < N$, one can show with high probability that the operator norm $\delta$ of the fluctuation matrix $\mathbf{\Delta}$ is less than 1.* [5]

See subsection C.3 for details. Then we can use the the above lemmata C.9 and C.11 to approximate the test error of KRR:

**Proposition C.13** (Bias Approximation). *Fix a sample $\mathbf{Z}$ of $\rho$ such that $\delta \overset{\text{def.}}{=} \|\mathbf{\Delta}\|_{op} < \frac{1}{2}$. Then the $bias_{test}$ term is bounded above and below by*

$$\left|bias - \left(\left\|\bar{\mathbf{P}}w\right\|_2^2 + \tilde{\gamma}_{>M}^2\right)\right| \leq 2\delta\left\|\bar{\mathbf{P}}w\right\|_2^2 + \|w\|_2^2\,\delta^2 p(\delta),$$

*where* $\bar{\mathbf{P}} \overset{\text{def.}}{=} \mathbf{\Lambda}(\mathbf{\Lambda} + \lambda\mathbf{I}_M)^{-1}$, $w = \lambda\mathbf{\Lambda}^{-1}\tilde{\gamma} - \tilde{\gamma}_{>M}E$, *and* $p(\delta) \overset{\text{def.}}{=} 5 + 4\delta + 4\delta^2$. *By writing* $E = (\eta_k)_{k=1}^M$, *the bounds simplify to*

$$bias \leq \tilde{\gamma}_{>M}^2 + (1 + 2\delta)\sum_{k=1}^{M}\frac{(\lambda\tilde{\gamma}_k - \tilde{\gamma}_{>M}\eta_k\lambda_k)^2}{(\lambda_k + \lambda)^2} + \|w\|_2^2\,\delta^2 p(\delta);$$

$$bias \geq \tilde{\gamma}_{>M}^2 + (1 - 2\delta)\sum_{k=1}^{M}\frac{(\lambda\tilde{\gamma}_k - \tilde{\gamma}_{>M}\eta_k\lambda_k)^2}{(\lambda_k + \lambda)^2} - \|w\|_2^2\,\delta^2 p(\delta).$$

---

[5]From there, we differentiate the approach from Bach [6]: From Propositions C.5 and C.7, it is inevitable to approximate the matrix $\mathbf{B}$, and we have $\mathbf{I}_M$ as support of the inverse. Bach instead uses RHKS basis to express the fluctuation matrix and is hence forced to use $\lambda\mathbf{I}_M$ as the support. As a result, he would need to require that the fluctuation is less than $\lambda$ and hence his requirement on $N$ is antiproportional to $\lambda$ in Theorem 5.1.

*Proof.* Let $w = \lambda \mathbf{\Lambda}^{-1} \tilde{\gamma} - \tilde{\gamma}_{>M} E$. We apply lemma C.4 followed by the 1st-order approximation $\mathbf{B}^{(1)}$ of the matrix $\mathbf{B}$ in lemma C.11:

$$
\begin{aligned}
\text{fitting error} = \|\mathbf{B}w\|_2^2 &= \left\| \mathbf{B}^{(1)}w + \left( \mathbf{B} - \mathbf{B}^{(1)} \right) w \right\|_2^2 \\
&= \left\| \mathbf{B}^{(1)}w \right\|_2^2 + w^\top \left( \mathbf{B} - \mathbf{B}^{(1)} \right) \mathbf{B}^{(1)}w + w^\top \mathbf{B}^{(1)} \left( \mathbf{B} - \mathbf{B}^{(1)} \right) w + \left\| \left( \mathbf{B} - \mathbf{B}^{(1)} \right) w \right\|_2^2 \\
&\le \left\| \mathbf{B}^{(1)}w \right\|_2^2 + 2 \left\| \mathbf{B}^{(1)} \right\|_{\mathrm{op}} \left\| \mathbf{B} - \mathbf{B}^{(1)} \right\|_{\mathrm{op}} \|w\|_2^2 + \left\| \mathbf{B} - \mathbf{B}^{(1)} \right\|_{\mathrm{op}}^2 \|w\|_2^2 \\
&\le \left\| \mathbf{B}^{(1)}w \right\|_2^2 + 2 \cdot (1+\delta) \cdot 2\delta^2 \|w\|_2^2 + 4\delta^4 \|w\|_2^2 \\
&\le \left\| \mathbf{B}^{(1)}w \right\|_2^2 + 4 \|w\|_2^2 \delta^2 (1 + \delta + \delta^2) \\
&\le \left\| \left( \mathbf{\bar{P}} - \mathbf{\bar{P}} \mathbf{\Delta} \mathbf{\bar{P}} \right) w \right\|_2^2 + 4 \|w\|_2^2 \delta^2 (1 + \delta + \delta^2) \\
&\le \left\| \mathbf{I}_M - \mathbf{\bar{P}} \mathbf{\Delta} \right\|_{\mathrm{op}}^2 \left\| \mathbf{\bar{P}}w \right\|_2^2 + 4 \|w\|_2^2 \delta^2 (1 + \delta + \delta^2) \\
&\le \left( 1 + 2 \left\| \mathbf{\bar{P}} \mathbf{\Delta} \right\|_{\mathrm{op}} + \left\| \mathbf{\bar{P}} \mathbf{\Delta} \right\|_{\mathrm{op}}^2 \right) \left\| \mathbf{\bar{P}}w \right\|_2^2 + 4 \|w\|_2^2 \delta^2 (1 + \delta + \delta^2) \\
&\le \left( 1 + 2 \left\| \mathbf{\bar{P}} \mathbf{\Delta} \right\|_{\mathrm{op}} \right) \left\| \mathbf{\bar{P}}w \right\|_2^2 + \|w\|_2^2 \delta^2 (5 + 4\delta + 4\delta^2) \\
&\le (1 + 2\delta) \left\| \mathbf{\bar{P}}w \right\|_2^2 + \|w\|_2^2 \delta^2 (5 + 4\delta + 4\delta^2).
\end{aligned}
$$

Hence we have the upper bound:

$$
\text{bias} \le \tilde{\gamma}_{>M}^2 + (1 + 2\delta) \left\| \mathbf{\bar{P}}w \right\|_2^2 + \|w\|_2^2 \delta^2 p(\delta).
$$

We argue similarly for the lower bound using: $\|\mathbf{A}\|_{\mathrm{op}} \|v\|_2^2 \ge v^\top \mathbf{A} v \ge - \|\mathbf{A}\|_{\mathrm{op}} \|v\|_2^2$ for any $\mathbf{A} \in \mathbb{R}^{M \times M}$, $v \in \mathbb{R}^{M \times 1}$. $\qquad\square$

We argue similarly for variance.

**Proposition C.14** (Variance Approximation). *Fix a sampling $\mathbf{Z}$ such that $\delta \overset{\text{def.}}{=} \|\mathbf{\Delta}\|_{op} < \frac{1}{2}$. Then we have*

$$
\left| variance - \frac{\sigma^2}{N} \sum_{k=1}^{M} \frac{\lambda_k^2}{(\lambda_k + \lambda)^2} \right| \le \delta \frac{\sigma^2}{N} \sum_{k=1}^{M} \frac{\lambda_k^2}{(\lambda_k + \lambda)^2} + M \frac{\sigma^2}{N} (1 + \delta) \delta^2 p(\delta),
$$

*where $p(\delta) \overset{\text{def.}}{=} 5 + 4\delta + 4\delta^2$, and $\sigma^2 \overset{\text{def.}}{=} \mathbb{E}[\epsilon^2]$ is the noise variance.*

*Proof.* Note that $\operatorname{Tr} \mathbf{A} \le M \|\mathbf{A}\|_{\mathrm{op}}$ for any matrix $\mathbf{A} \in \mathbb{R}^{M \times M}$. Since $\mathbf{B}^2(\mathbf{I}_M + \mathbf{\Delta}) = (\mathbf{B}^{(1)})^2(\mathbf{I}_M + \mathbf{\Delta}) + 2\mathbf{B}^{(1)} \left( \mathbf{B} - \mathbf{B}^{(1)} \right) (\mathbf{I}_M + \mathbf{\Delta}) + \left( \mathbf{B} - \mathbf{B}^{(1)} \right)^2 (\mathbf{I}_M + \mathbf{\Delta})$, we can bound the residue term by $\delta$:

$$
\begin{aligned}
\operatorname{Tr} &\left[ 2\mathbf{B}^{(1)} \left( \mathbf{B} - \mathbf{B}^{(1)} \right) (\mathbf{I}_M + \mathbf{\Delta}) + \left( \mathbf{B} - \mathbf{B}^{(1)} \right)^2 (\mathbf{I}_M + \mathbf{\Delta}) \right] \\
&\le M(1+\delta) \left\| \mathbf{B} - \mathbf{B}^{(1)} \right\|_{\mathrm{op}} \left( 2 \left\| \mathbf{B}^{(1)} \right\|_{\mathrm{op}} + \left\| \mathbf{B} - \mathbf{B}^{(1)} \right\|_{\mathrm{op}} \right) \\
&\le M(1+\delta) \cdot 2\delta^2 (2(1+\delta) + 2\delta^2) \\
&\le 4M\delta^2 (1+\delta)(1 + \delta + \delta^2),
\end{aligned}
$$

For the main terms, we have

$$
\begin{aligned}
\operatorname{Tr}[(\mathbf{B}^{(1)})^2 (\mathbf{I}_M + \mathbf{\Delta})] &\le \operatorname{Tr}[\mathbf{\bar{P}}^2] \cdot \left\| (\mathbf{I}_M - \mathbf{\Delta} \mathbf{\bar{P}})^2 (\mathbf{I}_M + \mathbf{\Delta}) \right\|_{\mathrm{op}} \\
&= \operatorname{Tr}[\mathbf{\bar{P}}^2] \left\| \mathbf{I}_M + \mathbf{\Delta}(\mathbf{I}_M - 2\mathbf{\bar{P}}) + (\mathbf{\Delta} \mathbf{\bar{P}})^2 - 2\mathbf{\Delta} \mathbf{\bar{P}} \mathbf{\Delta} + (\mathbf{\Delta} \mathbf{\bar{P}})^2 \mathbf{\Delta} \right\|_{\mathrm{op}} \\
&\le \operatorname{Tr}[\mathbf{\bar{P}}^2] \left\| \mathbf{I}_M + \mathbf{\Delta}(\mathbf{I}_M - 2\mathbf{\bar{P}}) \right\|_{\mathrm{op}} + M \left\| (\mathbf{\Delta} \mathbf{\bar{P}})^2 - 2\mathbf{\Delta} \mathbf{\bar{P}} \mathbf{\Delta} + (\mathbf{\Delta} \mathbf{\bar{P}})^2 \mathbf{\Delta} \right\|_{\mathrm{op}} \\
&\le \operatorname{Tr}[\mathbf{\bar{P}}^2](1+\delta) + M\delta^2(1+\delta).
\end{aligned}
$$

We apply Theorem C.7 to yield a bound on variance:

$$\left| \text{variance} - \frac{\sigma^2}{N} \sum_{k=1}^{M} \frac{\lambda_k^2}{(\lambda_k + \lambda)^2} \right| \leq \delta \frac{\sigma^2}{N} \sum_{k=1}^{M} \frac{\lambda_k^2}{(\lambda_k + \lambda)^2} + M \frac{\sigma^2}{N} (1 + \delta) \delta^2 p(\delta).$$

$\square$

Note that the above propositions C.13 and C.14 give absolute (non-probabilistic) bounds on the test error, once $\delta$ is controlled.

## C.3 Concentration Results

In this subsection, we focus on bounding the operator norm $\delta$ of the fluctuation matrix $\mathbf{\Delta}$.

First, we establish some concentration results.

**Lemma C.15** (Theorem 3.59 in [37]). *Let $\mathbf{A}$ be an $n \times N$ matrix with independent isotropic sub-Gaussian columns in $\mathbb{R}^n$ which sub-gaussian norm is bounded by a positive constant $G > 0$. Then for all $t \geq 0$, with probability at least $1 - 2\exp(-\frac{1}{3}t^2)$, we have*

$$\left\| \frac{1}{N} \mathbf{A}\mathbf{A}^\top - \mathbf{I}_n \right\|_{op} \leq \max(a, a^2), \tag{34}$$

*where $a \overset{\text{def.}}{=} C\sqrt{\frac{n}{N}} + \frac{t}{\sqrt{N}}$, for all constant $C \geq 12G^2$ .*

*Proof.* Let $a \overset{\text{def.}}{=} C\sqrt{\frac{n}{N}} + \frac{t}{\sqrt{N}}$ with $C > 0$ to be determined, and $\epsilon \overset{\text{def.}}{=} \max\{a, a^2\}$. The first step to show that :

$$\max_{x \in \mathcal{N}} \left| \frac{1}{N} \left\| \mathbf{A}^\top x \right\|_2^2 - 1 \right| \leq \epsilon$$

for some $\frac{1}{4}$-net $\mathcal{N}$ on the sphere $\mathbb{S}^{n-1} \subset \mathbb{R}^n$. Choose such a net $\mathcal{N}$ with $|\mathcal{N}| < \left(1 + \frac{2}{1/4}\right)^n = 9^n$.

Let $\mathbf{A}_i$ be the $i$th column of the matrix $\mathbf{A}$ and let $Z_i \overset{\text{def.}}{=} \mathbf{A}_i^\top x$ be a random variable. By definition of $\mathbf{A}$, $Z_i$ is centered with unit variance with sub-Gaussian norm upper bounded by $G$. Note that $G \geq \frac{1}{\sqrt{2}}\mathbb{E}[Z_i^2]^{1/2} = \frac{1}{\sqrt{2}}$, and the random variable $Z_i^2 - 1$ is centered and has sub-exponential norm upper bounded by $4G^2$. Hence by an exponential deviation inquality [6], we have, for any $x \in \mathbb{S}^{n-1}$:

$$\mathbb{P}\left\{ \left| \frac{1}{N} \left\| \mathbf{A}^\top x \right\|_2^2 - 1 \right| \geq \frac{\epsilon}{2} \right\} = \mathbb{P}\left\{ \left| \frac{1}{N} \sum_{i=1}^{N} Z_i^2 - 1 \right| \geq \frac{\epsilon}{2} \right\}$$

$$\leq 2\exp\left( -\frac{1}{2}e^{-1}G^{-4} \min\{\epsilon, \epsilon^2\} \right)$$

$$= \leq 2\exp\left( -\frac{1}{2}e^{-1}G^{-4}a^2 \right)$$

$$\leq 2\exp\left( -\frac{1}{2}e^{-1}G^{-4}(C^2 n + t^2) \right).$$

Then by union bound, we have

$$\mathbb{P}\left\{ \max_{x \in \mathcal{N}} \left| \frac{1}{N} \left\| \mathbf{A}^\top x \right\|_2^2 - 1 \right| \geq \frac{\epsilon}{2} \right\} \leq 9^n \cdot 2\exp\left( -\frac{1}{2}e^{-1}G^{-4}(C^2 n + t^2) \right)$$

$$\leq 2\exp\left( -\frac{1}{2}e^{-1}G^{-4}t^2 \right),$$

---

[6]This inequality is Corollary 5.17 from [37].

for $C \geq \sqrt{2e \log 9} G^2$. Since $12 > \sqrt{2e \log 9}$, for simplicity, we assume $C > 12G^2$. Moreover, since $G \geq \frac{1}{\sqrt{2}}$, we have $\frac{1}{2} e^{-1} G^{-4} \leq \frac{1}{3}$, we have

$$\mathbb{P}\left\{ \max_{x \in \mathcal{N}} \left| \frac{1}{N} \left\| \mathbf{A}^\top x \right\|_2^2 - 1 \right| \geq \frac{\epsilon}{2} \right\} \leq 2 \exp\left( -\frac{1}{3} t^2 \right).$$

Then by the $\frac{1}{4}$-net argument, with probability at least $1 - 2\exp\left(-\frac{1}{3} t^2\right)$, we have

$$\left\| \frac{1}{N} \mathbf{A}\mathbf{A}^\top - \mathbf{I}_n \right\|_{op} \leq \frac{4}{2} \max_{x \in \mathcal{N}} \left| \frac{1}{N} \left\| \mathbf{A}^\top x \right\|_2^2 - 1 \right|$$

$$\leq \epsilon = \max\{a, a^2\}.$$

$\square$

**Lemma C.16.** *Assume Assumption 4.1 holds and that $N > \exp(4(12G^2)^2(M+1))$. Then with a probability of at least $1 - 2/N$, we have*

$$\max\{\delta, \|\boldsymbol{E}_M\|_2\} \leq \sqrt{\frac{\log N}{N}}.$$

*Proof.* Set $n = M + 1$, $\mathbf{A} = \binom{\boldsymbol{\Psi}_{\leq M}}{\psi_{>M}(\mathbf{X})^\top} \in \mathbb{R}^{(M+1) \times N}$. Then

$$\frac{1}{N} \mathbf{A}\mathbf{A}^\top - \mathbf{I}_n = \begin{pmatrix} \frac{1}{N} \boldsymbol{\Psi}_{\leq M} \boldsymbol{\Psi}_{\leq M}^\top & \boldsymbol{E}_M \\ \boldsymbol{E}_M^\top & \eta_{>M} + 1 \end{pmatrix} - \mathbf{I}_n = \begin{pmatrix} \boldsymbol{\Delta}_M & \boldsymbol{E}_M \\ \boldsymbol{E}_M^\top & \eta_{>M} \end{pmatrix}.$$

where $\eta_{>M} \overset{\text{def.}}{=} \frac{1}{N} \sum_{i=1}^N \psi_{>M}(x_i)^2 - 1$. On one hand, the operator norm of the above matrix bounds $\delta$ and $\|\boldsymbol{E}_M\|_2$ from above:

$$\left\| \begin{pmatrix} \boldsymbol{\Delta}_M & \boldsymbol{E}_M \\ \boldsymbol{E}_M^\top & \eta_{>M} \end{pmatrix} \right\|_{op} = \max_{\|\mathbf{u}\|_2^2 + v^2 = 1} \left\| \begin{pmatrix} \boldsymbol{\Delta}_M & \boldsymbol{E}_M \\ \boldsymbol{E}_M^\top & \eta_{>M} \end{pmatrix} \begin{pmatrix} \mathbf{u} \\ v \end{pmatrix} \right\|_2$$

$$= \max_{\|\mathbf{u}\|_2^2 + v^2 = 1} \left\| \begin{pmatrix} \boldsymbol{\Delta}_M \mathbf{u} + v\boldsymbol{E}_M \\ \boldsymbol{E}_M^\top \mathbf{u} + \eta_{>M} v \end{pmatrix} \right\|_2$$

$$\geq \max_{\|\mathbf{u}\|_2^2 + v^2 = 1} \|\boldsymbol{\Delta}_M \mathbf{u} + v\boldsymbol{E}_M\|_2$$

$$\geq \max_{\|\mathbf{u}\|_2^2 = 1, v = 0} \|\boldsymbol{\Delta}_M \mathbf{u} + v\boldsymbol{E}_M\|_2$$

$$\geq \max_{\|\mathbf{u}\|_2^2 = 1} \|\boldsymbol{\Delta}_M \mathbf{u}\|_2 = \delta,$$

and

$$\left\| \begin{pmatrix} \boldsymbol{\Delta}_M & \boldsymbol{E}_M \\ \boldsymbol{E}_M^\top & \eta_{>M} \end{pmatrix} \right\|_{op} \geq \max_{\|\mathbf{u}\|_2^2 + v^2 = 1} \|\boldsymbol{\Delta}_M \mathbf{u} + v\boldsymbol{E}_M\|_2 \geq \max_{\|\mathbf{u}\|_2^2 = 0, |v| = 1} \|\boldsymbol{\Delta}_M \mathbf{u} + v\boldsymbol{E}_M\|_2 = \|\boldsymbol{E}_M\|_2.$$

On the other hand, set $t = \frac{1}{2}\sqrt{\log N}$, $C = 12G^2$, since $N > \exp(4C^2(M+1))$, we have

$$a = C\sqrt{\frac{n}{N}} + \frac{t}{\sqrt{N}} = 12G^2 \sqrt{\frac{M+1}{N}} + \frac{1}{2}\sqrt{\frac{\log N}{N}} \leq \sqrt{\frac{\log N}{N}} < 1.$$

By Lemma C.16, then with probability of at least $1 - 2\exp(-\frac{1}{3}t^2) = 1 - 2\exp(-\frac{1}{12})/N > 1 - 2/N$, we have

$$\left\| \begin{pmatrix} \boldsymbol{\Delta}_M & \boldsymbol{E}_M \\ \boldsymbol{E}_M^\top & \eta_{>M} \end{pmatrix} \right\|_{op} \leq \max\{a, a^2\} = a \leq \sqrt{\frac{\log N}{N}}.$$

Combine the both results and we conclude the upper bounds. $\square$

In particular, as $N \to \infty$, $\delta$ vanishes almost surely. In empirical calculation, if the requirement $N > \exp(4(12G^2)^2(M+1))$ exponential in $M$ is too demanding for a large integer $M$, we can take $t = N^s$ for any positive number $s \in \left(0, \frac{1}{2}\right)$ instead of $t = \frac{1}{2}\log N$. In this way, we decrease the requirement to $N$ polynomial in $M$ in sacrificing the decay from $\mathcal{O}\left(\sqrt{\frac{\log N}{N}}\right)$ to $\mathcal{O}\left(N^{s-1/2}\right)$. For simplicity purpose, we do not list out the result with this decay in this paper.

## C.4 Refined Test Error Analysis

We can apply the above concentration results to refine the following bounds on the finite-rank KRR test error. First of all, we realize the decay of target function coefficient comparable to the spectral decay:

**Definition C.17** (Comparable Decay). *Denote* $\underline{r} \overset{\text{def.}}{=} \min_k\{|\tilde{\gamma}_k/\lambda_k|\}$ *and* $\overline{r} \overset{\text{def.}}{=} \max_k\{|\tilde{\gamma}_k/\lambda_k|\}$.

### C.4.1 Refined Bounds on Bias

Recall that Proposition C.13 bounding the bias in terms of $\delta$ and $\eta_k$. For the former one, we can choose: for $N > \max\{\exp(4(12G^2)^2(M+1)), 9\}$, by Lemma C.16, with probability of at least $1 - 2/N$, we have $\delta \leq \sqrt{\frac{\log N}{N}} < \sqrt{\frac{\log 9}{9}} < \frac{1}{2}$. For the latter one, we have to control the vector $w$:

**Lemma C.18.** *Let* $w = \lambda\mathbf{\Lambda}^{-1}\tilde{\boldsymbol{\gamma}} - \tilde{\gamma}_{>M}\boldsymbol{E}$. *We have*

$$\|w\|_2^2 \leq \left(\lambda\overline{r}\sqrt{M} + |\tilde{\gamma}_{>M}|\,\|\boldsymbol{E}\|_2\right)^2;$$

$$\frac{\lambda^2\lambda_M}{(\lambda_M+\lambda)^2}\|\tilde{f}_{\leq M}\|_{\mathcal{H}}^2 - \frac{1}{2}|\tilde{\gamma}_{>M}|\|\tilde{f}_{\leq M}\|_{L_\rho^2}\|\boldsymbol{E}\|_2 \leq \|\bar{\mathbf{P}}w\|_2^2 \leq \lambda\|\tilde{f}_{\leq M}\|_{\mathcal{H}}^2 + \frac{1}{2}|\tilde{\gamma}_{>M}|\|\tilde{f}_{\leq M}\|_{L_\rho^2}\|\boldsymbol{E}\|_2 + \tilde{\gamma}_{>M}^2\|\boldsymbol{E}\|_2^2.$$

*Proof.* Since $\lambda^2\underline{r}^2M \leq \left\|\lambda\mathbf{\Lambda}^{-1}\tilde{\boldsymbol{\gamma}}\right\|_2^2 \leq \lambda^2\overline{r}^2M$ and $\|\tilde{\gamma}_{>M}\boldsymbol{E}\|_2^2 = \tilde{\gamma}_{>M}^2\|\boldsymbol{E}\|_2^2$, we have

$$\|w\|_2^2 \leq \left(\lambda\overline{r}\sqrt{M} + |\tilde{\gamma}_{>M}|\,\|E\|_2\right)^2.$$

Similarly, we can bound $\|\bar{\mathbf{P}}w\|_2$. Observe that:

$$\|\bar{\mathbf{P}}w\|_2^2 = \underbrace{\lambda^2\sum_{k=1}^{M}\frac{\tilde{\gamma}_k^2}{(\lambda_k+\lambda)^2}}_{I} \underbrace{-2\lambda\tilde{\gamma}_{>M}\sum_{k=1}^{M}\frac{\tilde{\gamma}_k\lambda_k\eta_k}{(\lambda_k+\lambda)^2}}_{II} + \underbrace{\tilde{\gamma}_{>M}^2\sum_{k=1}^{M}\frac{\lambda_k^2\eta_k^2}{(\lambda_k+\lambda)^2}}_{III}.$$

Since $1 \geq \frac{\lambda}{\lambda_k+\lambda} \geq \frac{\lambda}{\lambda_M+\lambda}$, we have the upper bound:

$$I = \lambda^2\sum_{k=1}^{M}\frac{\tilde{\gamma}_k^2}{(\lambda_k+\lambda)^2} \leq \lambda\sum_{k=1}^{M}\frac{\lambda}{\lambda_k+\lambda}\frac{\tilde{\gamma}_k^2}{\lambda_k+\lambda} \leq \lambda\sum_{k=1}^{M}\frac{\tilde{\gamma}^2}{\lambda_k} = \lambda\|\tilde{f}_{\leq M}\|_{\mathcal{H}}^2. \tag{35}$$

where $\tilde{f}_{\leq M} \overset{\text{def.}}{=} \sum_{k=1}^{M}\tilde{\gamma}_k\psi_k = \tilde{f} - \tilde{\gamma}_{>M}\psi_{>M}$. For the lower bound, we have:

$$I = \lambda^2\sum_{k=1}^{M}\frac{\tilde{\gamma}_k^2}{(\lambda_k+\lambda)^2} \geq \lambda^2\sum_{k=1}^{M}\frac{\lambda_k}{(\lambda_k+\lambda)^2}\frac{\tilde{\gamma}_k^2}{\lambda_k} \geq \lambda^2\frac{\lambda_M}{(\lambda_M+\lambda)^2}\|\tilde{f}_{\leq M}\|_{\mathcal{H}}^2 \tag{36}$$

Similarly, since $4\lambda\lambda_k \leq (\lambda_k+\lambda)^2$,

$$|II| = 2\lambda|\tilde{\gamma}_{>M}|\sum_{k=1}^{M}\frac{|\tilde{\gamma}_k|\lambda_k|\eta_k|}{(\lambda_k+\lambda)^2} \leq \frac{1}{2}|\tilde{\gamma}_{>M}|\sum_{k+1}^{M}|\tilde{\gamma}_k\eta_k| \leq \frac{1}{2}|\tilde{\gamma}_{>M}|\sqrt{\sum_{k+1}^{M}\tilde{\gamma}_k^2\sum_{k=1}^{M}\eta_k^2} \leq \frac{1}{2}|\tilde{\gamma}_{>M}|\|\tilde{f}_{\leq M}\|_{L_\rho^2}\|\boldsymbol{E}\|_2.$$

And

$$III = \tilde{\gamma}_{>M}^2\sum_{k=1}^{M}\frac{\lambda_k^2\eta_k^2}{(\lambda_k+\lambda)^2} \leq \tilde{\gamma}_{>M}^2\sum_{k=1}^{M}\eta_k^2 = \tilde{\gamma}_{>M}^2\|\boldsymbol{E}\|_2^2.$$

$\square$

Combining the above result, we state the following theorem:

**Theorem C.19.** *For* $N > \max\left\{\exp(4(12G^2)^2(M+1)), 9\right\}$ *and for any constant* $C_1 > 8\left(\lambda\bar{r}\sqrt{M} + \frac{1}{2}|\tilde{\gamma}_{>M}|\right)^2 + \frac{5}{2}\|\tilde{f}\|_{L^2_\rho}^2$ *(independent to* $N$*), with a probability of at least* $1 - 2/N$, *we have the upper and lower bounds of bias:*

$$bias \leq \tilde{\gamma}_{>M}^2 + \lambda\|\tilde{f}_{\leq M}\|_{\mathcal{H}}^2 + \left(\frac{1}{4}\|\tilde{f}\|_{L^2_\rho}^2 + 2\lambda\|\tilde{f}_{\leq M}\|_{\mathcal{H}}^2\right)\sqrt{\frac{\log N}{N}} + C_1\frac{\log N}{N};$$

$$bias \geq \tilde{\gamma}_{>M}^2 + \frac{\lambda^2\lambda_M}{(\lambda_M + \lambda)^2}\|\tilde{f}_{\leq M}\|_{\mathcal{H}}^2 - \left(\frac{1}{4}\|\tilde{f}\|_{L^2_\rho}^2 + \frac{2\lambda^2}{\lambda_1 + \lambda}\|\tilde{f}_{\leq M}\|_{\mathcal{H}}^2\right)\sqrt{\frac{\log N}{N}} - C_1\frac{\log N}{N}.$$

*For* $\lambda \to 0$, *we have a simpler bound: with a probability of at least* $1 - 2/N$, *we have*

$$\lim_{\lambda\to 0} bias \leq \left(1 + \frac{\log N}{N}\right)\tilde{\gamma}_{>M}^2 + 6\tilde{\gamma}_{>M}^2\left(\frac{\log N}{N}\right)^{\frac{3}{2}};$$

$$\lim_{\lambda\to 0} bias \geq \left(1 - \frac{\log N}{N}\right)\tilde{\gamma}_{>M}^2 - 6\tilde{\gamma}_{>M}^2\left(\frac{\log N}{N}\right)^{\frac{3}{2}}.$$

(37)

*For* $\tilde{\gamma}_{>M}^2 = 0$, *that is* $\tilde{f} \in \mathcal{H}$, *we have a simpler upper bound on bias: with a probability of at least* $1 - 2/N$, *we have*

$$bias \leq \lambda\|\tilde{f}\|_{\mathcal{H}}^2\left(1 + 2\sqrt{\frac{\log N}{N}}\right) + C_1\frac{\log N}{N};$$

$$bias \geq \frac{\lambda^2\lambda_M}{(\lambda_M + \lambda)^2}\|\tilde{f}\|_{\mathcal{H}}^2\left(1 - 2\sqrt{\frac{\log N}{N}}\right) - C_1\frac{\log N}{N}.$$

(38)

*Proof.* By Proposition C.13 and Lemma C.18,

$$\text{fitting error} \leq (1 + 2\delta)\left\|\bar{\mathbf{P}}w\right\|_2^2 + \|w\|_2^2\,\delta^2 p(\delta) \tag{39}$$

$$\leq (1 + 2\delta)(\lambda\|\tilde{f}_{\leq M}\|_{\mathcal{H}}^2 + \frac{1}{2}|\tilde{\gamma}_{>M}|\|\tilde{f}_{\leq M}\|_{L^2_\rho}\|\mathbf{E}\|_2 + \tilde{\gamma}_{>M}^2\|\mathbf{E}\|_2^2) + \|w\|_2^2\,\delta^2 p(\delta) \tag{40}$$

$$\leq (1 + 2\delta)(\lambda\|\tilde{f}_{\leq M}\|_{\mathcal{H}}^2 + \frac{1}{4}\|\tilde{f}\|_{L^2_\rho}^2\|\mathbf{E}\|_2 + \tilde{\gamma}_{>M}^2\|\mathbf{E}\|_2^2) + \|w\|_2^2\,\delta^2 p(\delta). \tag{41}$$

where in line (39), we use Proposition C.13; in line (40), we use Lemma C.18; in line (40), we use the fact that $2ab \leq a^2 + b^2$ where $a = |\tilde{\gamma}_{>M}|, b = \|\tilde{f}_{\leq M}\|_{L^2_\rho}$.

Now we apply the concentration result in Lemma C.16: with a probability of at least $1 - 2/N$:

$$\text{fitting error} \leq \left(1 + 2\sqrt{\frac{\log N}{N}}\right)\lambda\|\tilde{f}_{\leq M}\|_{\mathcal{H}}^2 + \frac{1}{4}\|\tilde{f}\|_{L^2_\rho}^2\sqrt{\frac{\log N}{N}}$$

$$+ \frac{\log N}{N}\left(\|w\|_2^2\,p(\delta) + (1 + 2\delta)\tilde{\gamma}_{>M}^2 + \frac{1}{2}\|\tilde{f}\|_{L^2_\rho}^2\right)$$

$$\leq \lambda\|\tilde{f}_{\leq M}\|_{\mathcal{H}}^2 + \left(\frac{1}{4}\|\tilde{f}\|_{L^2_\rho}^2 + 2\lambda\|\tilde{f}_{\leq M}\|_{\mathcal{H}}^2\right)\sqrt{\frac{\log N}{N}} + C_1\frac{\log N}{N},$$

where we choose $C_1 > 0$ to be such that:

$$\|w\|_2^2\,p(\delta) + (1 + 2\delta)\tilde{\gamma}_{>M}^2 + \frac{1}{2}\|\tilde{f}\|_{L^2_\rho}^2 \leq \|w\|_2^2\,p(\delta) + \left(1 + 2\delta + \frac{1}{2}\right)\|\tilde{f}\|_{L^2_\rho}^2$$

$$\leq \|w\|_2^2\,p\left(\frac{1}{2}\right) + \left(1 + 2\cdot\frac{1}{2} + \frac{1}{2}\right)\|\tilde{f}\|_{L^2_\rho}^2$$

$$\leq 8\left(\lambda\bar{r}\sqrt{M} + |\tilde{\gamma}_{>M}|\,\|E\|_2\right)^2 + \frac{5}{2}\|\tilde{f}\|_{L^2_\rho}^2$$

$$\leq 8\left(\lambda\bar{r}\sqrt{M} + \frac{1}{2}|\tilde{\gamma}_{>M}|\right)^2 + \frac{5}{2}\|\tilde{f}\|_{L^2_\rho}^2 < C_1.$$

Hence we have an upper bound for the bias. We argue similarly for the lower bound:

$$\text{fitting error} \geq \left(1 - 2\sqrt{\frac{\log N}{N}}\right)\frac{\lambda^2 \lambda_M}{(\lambda_M + \lambda)^2}\|\tilde{f}_{\leq M}\|_{\mathcal{H}}^2 - \frac{1}{4}\|\tilde{f}\|_{L_\rho^2}^2\sqrt{\frac{\log N}{N}}$$

$$- \frac{\log N}{N}\left(\|w\|_2^2 p(\delta) + (1 + 2\delta)\tilde{\gamma}_{>M}^2 + |\tilde{\gamma}_{>M}|\|\tilde{f}_{\leq M}\|_{L_\rho^2}\right)$$

$$\geq \frac{\lambda^2 \lambda_M}{(\lambda_M + \lambda)^2}\|\tilde{f}_{\leq M}\|_{\mathcal{H}}^2 - \left(\frac{1}{4}\|\tilde{f}\|_{L_\rho^2}^2 + \frac{2\lambda^2}{\lambda_1 + \lambda}\|\tilde{f}_{\leq M}\|_{\mathcal{H}}^2\right)\sqrt{\frac{\log N}{N}} - C_1\frac{\log N}{N}.$$

For $\lambda \to 0$, note that $w \to -\tilde{\gamma}_{>M}\boldsymbol{E}$. This yields

$$\lim_{\lambda \to 0}\text{fitting error} \leq \lim_{\lambda \to 0}\left\{(1 + 2\delta)\left\|\bar{\mathbf{P}}w\right\|_2^2 + \|w\|_2^2\delta^2 p(\delta)\right\}$$

$$= (1 + 2\delta)\left\|-\tilde{\gamma}_{>M}\boldsymbol{E}\right\|_2^2 + \left\|-\tilde{\gamma}_{>M}\boldsymbol{E}\right\|_2^2\delta^2 p(\delta)$$

$$= \tilde{\gamma}_{>M}^2\left\|\boldsymbol{E}\right\|_2^2(1 + \delta(2 + \delta p(\delta))).$$

Hence, by plugging in $\delta < \frac{1}{2}$, with probability of at least $1 - 2/N$,

$$\lim_{\lambda \to 0}\text{fitting error} \leq \tilde{\gamma}_{>M}^2\frac{\log N}{N}\left(1 + 6\sqrt{\frac{\log N}{N}}\right)$$

$$\lim_{\lambda \to 0}\text{bias} \leq \left(1 + \frac{\log N}{N}\right)\tilde{\gamma}_{>M}^2 + 6\tilde{\gamma}_{>M}^2\left(\frac{\log N}{N}\right)^{\frac{3}{2}}.$$

For lower bound, it follows similarly:

$$\lim_{\lambda \to 0}\text{bias} \geq \left(1 - \frac{\log N}{N}\right)\tilde{\gamma}_{>M}^2 - 6\tilde{\gamma}_{>M}^2\left(\frac{\log N}{N}\right)^{\frac{3}{2}},$$

and we obtain line (37). For the case where $\tilde{\gamma}_{>M} = 0$, recalculate and simplify line (40) to obtain line (38). $\qquad\square$

### C.4.2  Refined Bounds on Variance

Similarly, we can refine Theorem C.14 to get a bound on the variance:

**Theorem C.20.** *For $N > \max\left\{(12G)^4(M + 1)^2, 9\right\}$, and set $C_2 = 12$ (independent to $N$), with a probability of at least $1 - 2/N$, we have the upper and lower bounds of variance:*

$$\text{variance} \leq \sigma^2\frac{M}{N}\left(1 + \sqrt{\frac{\log N}{N}} + C_2\frac{\log N}{N}\right);$$

$$\text{variance} \geq \frac{\lambda_M^2}{(\lambda_M + \lambda)^2}\sigma^2\frac{M}{N}\left(1 - \sqrt{\frac{\log N}{N}}\right) - C_2\sigma^2\frac{M}{N}\frac{\log N}{N}.$$

*Proof.* We argue analogously as in Theorem C.19: by Proposition C.14 and Lemma C.16, we have

$$\text{variance} \leq (1 + \delta)\frac{\sigma^2}{N}\sum_{k=1}^{M}\frac{\lambda_k^2}{(\lambda_k + \lambda)^2} + M\frac{\sigma^2}{N}(1 + \delta)\delta^2 p(\delta)$$

$$\leq (1 + \delta)\sigma^2\frac{M}{N} + \sigma^2\frac{M}{N}(1 + \delta)\delta^2 p(\delta)$$

$$\leq \left(1 + \sqrt{\frac{\log N}{N}}\right)\sigma^2\frac{M}{N} + \sigma^2\frac{M}{N}\frac{\log N}{N}\left(1 + \frac{1}{2}\right)p\left(\frac{1}{2}\right)$$

$$\leq \left(1 + \sqrt{\frac{\log N}{N}}\right)\sigma^2\frac{M}{N} + 12\sigma^2\frac{M}{N}\frac{\log N}{N}$$

Hence we can choose $C_2 = 12$. For the lower bound, since $\frac{\lambda_k^2}{(\lambda_k+\lambda)^2} > \frac{\lambda_M^2}{(\lambda_M+\lambda)^2}$, we have

$$\text{variance} \geq (1-\delta)\frac{\sigma^2}{N}\sum_{k=1}^{M}\frac{\lambda_k^2}{(\lambda_k+\lambda)^2} - M\frac{\sigma^2}{N}(1+\delta)\delta^2 p(\delta)$$

$$\geq \frac{\lambda_M^2}{(\lambda_M+\lambda)^2}\sigma^2\frac{M}{N}\left(1 - \sqrt{\frac{\log N}{N}}\right) - 12\sigma^2\frac{M}{N}\frac{\log N}{N}.$$

$\square$

Note that in both Theorems C.19 and C.20, the constants $C_1, C_2 > 0$ is not optimized.

# D   Numerical Validation

In this section, we illustrate our result for KRR with two different finite rank kernels.

## D.1   Truncated NTK

First, we need to define a finite-rank kernel $K : \mathcal{X} \times \mathcal{X} \to \mathbb{R}$. We set $\mathcal{X} = \mathbb{S}^1 \subset \mathbb{R}^2$. By reparametrization, we write $\mathbb{S}^1 \cong [0, 2\pi]/_{0\sim 2\pi}$. We assume the data are drawn uniformly on the circle, that is $\rho_{\mathcal{X}} = \text{unif}[\mathbb{S}^1]$. We can use the Fourier functions $\cos(k\cdot), \sin(k\cdot)$ as the orthogonal eigenfunctions of the kernel. We define the NTK

$$K^{(\infty)}(\theta, \theta') \stackrel{\text{def.}}{=} \frac{\cos(\theta-\theta')\,(\pi - |\theta-\theta'|)}{2\pi}$$

for all $\theta, \theta' \in [0, 2\pi]$. 3) We choose a rank-$M$ truncation $K(\theta, \theta') = \sum_{k=1}^{M}\lambda_k\psi_k(\theta)\psi_k(\theta')$ for all $\theta, \theta' \in [0, 2\pi]$. For the first few eigenvalues of the kernel, please see Table 3 for example. Before

| $k$ | 1 | 2 | 3 | 4 | 5 | 6 | 7 | $\infty$ |
|---|---|---|---|---|---|---|---|---|
| $\lambda_k$ | $\frac{1}{\pi^2}$ | $\frac{1}{8}$ | $\frac{1}{8}$ | $\frac{5}{9\pi^2}$ | $\frac{5}{9\pi^2}$ | $\frac{17}{225\pi^2}$ | $\frac{17}{225\pi^2}$ | - |
| $\psi_k(\theta)$ | 1 | $\sqrt{2}\cos(\theta)$ | $\sqrt{2}\sin(\theta)$ | $\sqrt{2}\cos(2\theta)$ | $\sqrt{2}\sin(2\theta)$ | $\sqrt{2}\cos(4\theta)$ | $\sqrt{2}\sin(4\theta)$ | - |
| $\sum_{k'=0}^{k}\lambda_{k'}$ | 0.1013 | 0.2263 | 0.3513 | 0.4076 | 0.4639 | 0.4716 | 0.4792 | 0.5 |

Table 3: The first few eigenvalues of the NTK

proceeding to test error computation, we present a training example, Figure 4, to give readers an intuition on the truncated NTK (tNTK).

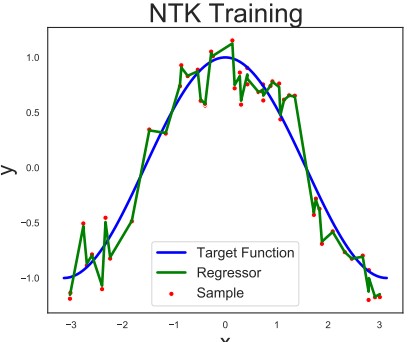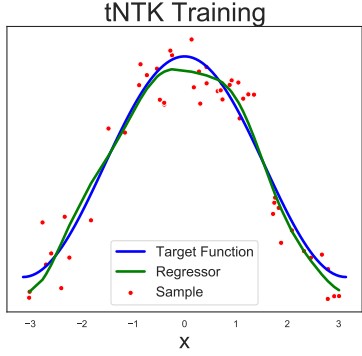

Figure 4:   (left): NTK training; (right): tNTK training where $N = 50, M = 7$. $\sigma^2 = 0.05, \lambda = \sigma^2/N$.

## D.2 Test Error Computations

In the following tNTK training, we set the hyperparameters as follows:

**Target function** We choose a simple target function $\tilde{f}(x) = \cos x = \frac{1}{\sqrt{2}} \psi_2(x)$. Throughout the experiment, we set the noise variance $\sigma^2 = 0.05$.

**Ridge** We choose $\lambda = \frac{\sigma^2}{N}$. In Figure 5 (left), we set $N = 50, \lambda = 0.05/50$ for tNTK training; (right) we set set $\lambda = 0.05/50$ for varying $N$ from 10 to 200.

**Error bars** In Figure 7 (right), for each value of $N$, we run over 10 iterations of random samples and compute the test error. The error bars are shown as the difference between the upper and the lower quartiles.

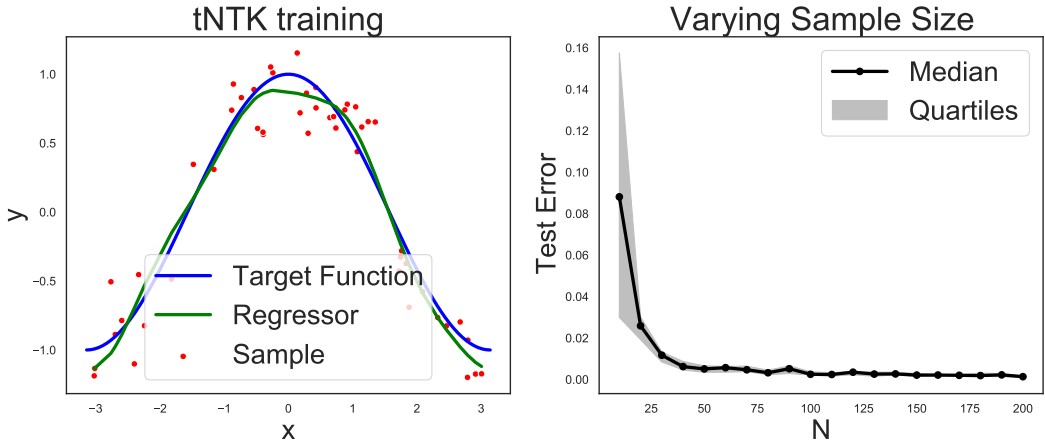

Figure 5: (left): tNTK training; (right): the decay of test error as $N$ varies.

**Lower bound** See the subsection below.

## D.3 Bound Comparison

We continue with the experiment on the tNTK this time with varying $N$ and compare our upper bound with [6].

**Upper bounds** In Figure 6, the expression of Bach's and our upper bounds are directly computed:

$$\text{Bach's upper bound} = 4\lambda \|\tilde{f}\|_{\mathcal{H}}^2 + \frac{8\sigma^2 R^2}{\lambda N}(1 + 2\log N)$$

$$\text{Our upper bound without residue} = \lambda \|\tilde{f}\|_{\mathcal{H}}^2 \left(1 + 2\sqrt{\frac{\log N}{N}}\right),$$

where the constants $\|\tilde{f}\|_{\mathcal{H}}^2$ and $R^2$ can be computed directed from the choice of kernel and target function. For simplicity reason, we drop the residue term $C_1 \frac{\log N}{N}$ since it is overshadowed by the other terms and the constant $C_1$ is not optimized.

## D.4 Legendre Kernel

To illustrate the bounds with another finite-rank, we choose a simple legendre kernel (LK):

$$K(x, z) = \sum_{k=0}^{M} \lambda_k P_k(x) P_k(z)$$

where $P_k$ is the Legendre polynomial of degree $k$, and $\lambda_k > 0$ are the eigenvalues.

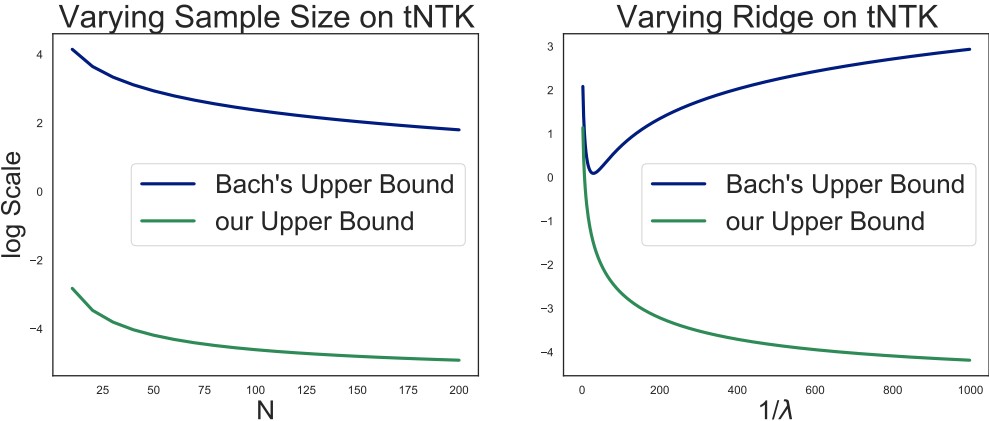

Figure 6: Test error bound improvement on tNTK. Same as Figure 2.

**Eigenvalues** To better compare the Legendre kernel $K$ with the NTK, we choose $\lambda_k = C \cdot (k+1)^{-2}$ of quadratic decay such that the spectral sums are the same: $\sum_{k=0}^{\infty} \lambda_k = 0.5$. Hence we choose $C = 0.5 / \sum_{k=1}^{\infty} k^{-2} = \frac{3}{\pi^2}$.

**Target function** We choose a simple target function $\tilde{f}(x) = x^2 = \frac{1}{3} P_0(x) + \frac{2}{3} P_2(x)$. Throughout the experiment, we set the noise variance $\sigma^2 = 0.05$.

### D.5 Test Error Computation

**Ridge** As before, our bound suggests that, to balance the bias and the variance with a fixed $N$, we can choose $\lambda = \frac{\sigma^2}{N}$. In Figure 7 (left), we set $N = 50, \lambda = 0.05/50$ for KRR training; (right) we set set $\lambda = 0.05/50$ for varying $N$ from 10 to 200.

**Error bars** In Figure 7 (right), for each value of $N$, we run over 10 iterations of random samples and compute the test error. The error bars are shown as the different between the upper and the lower quartiles. The median is taken as average.

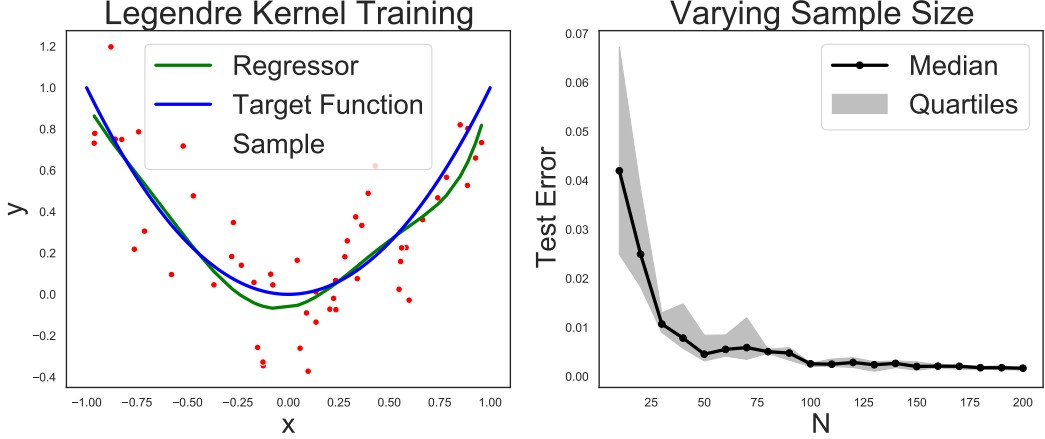

Figure 7: (left): LK training; (right): the decay of test error as $N$ varies. Same as Figure 1.

**Upper bounds** In Figure 8, the expression of Bach's and our upper bounds are directly computed:

$$\text{Bach's upper bound} = 4\lambda\|\tilde{f}\|_{\mathcal{H}}^2 + \frac{8\sigma^2 R^2}{\lambda N}(1 + 2\log N)$$

$$\text{Our upper bound without residue} = \lambda\|\tilde{f}\|_{\mathcal{H}}^2 \left(1 + 2\sqrt{\frac{\log N}{N}}\right),$$

where the constants $\|\tilde{f}\|_{\mathcal{H}}^2$ and $R^2$ can be computed directed from the choice of kernel and target function.

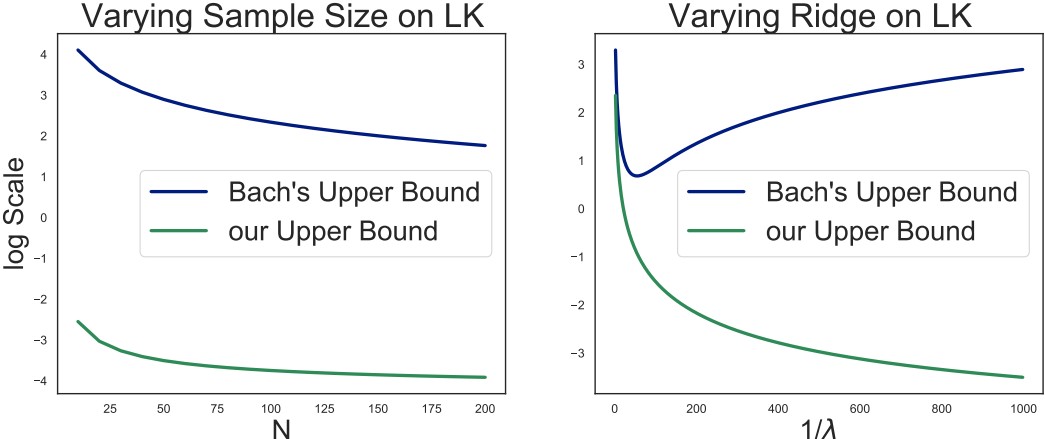

Figure 8: Test error bound improvement on LK. Same as Figure 2.

**Lower Bound** Last but not least, we need to show our lower bound is valid. To see this clearly, we need to write the bound in exact sums instead of in HKRS norm square $\|\tilde{f}\|_{\mathcal{H}}^2$: namely, we compute $I$

$$\frac{\lambda^2\lambda_M}{(\lambda_M+\lambda)^2}\|\tilde{f}\|_{\mathcal{H}}^2 \leq I = \lambda^2\sum_{k=1}^{M}\frac{\tilde{\gamma}_k^2}{(\lambda_k+\lambda)^2} \leq \lambda\|\tilde{f}\|_{\mathcal{H}}^2, \tag{42}$$

instead of using the inequality (42) in Lemma C.18; and

$$M\frac{\lambda_M^2}{(\lambda_M+\lambda)^2} \leq \sum_{k=1}^{M}\frac{\lambda_k^2}{(\lambda_k+\lambda)^2} \leq M, \tag{43}$$

instead of using the inequality (42) in Theorem C.20. Then we can compute our bounds as:

$$\text{Our upper bound without residue} = \lambda^2 I\left(1 + 2\sqrt{\frac{\log N}{N}}\right) + \frac{\sigma^2}{N}\sum_{k=1}^{M}\frac{\lambda_k^2}{(\lambda_k+\lambda)^2}\left(1 + \sqrt{\frac{\log N}{N}}\right),$$

$$\text{Our lower bound without residue} = \lambda^2 I\left(1 - 2\sqrt{\frac{\log N}{N}}\right) + \frac{\sigma^2}{N}\sum_{k=1}^{M}\frac{\lambda_k^2}{(\lambda_k+\lambda)^2}\left(1 - \sqrt{\frac{\log N}{N}}\right),$$

and we drop the residue terms $C_1\frac{\log N}{N}$ and $C_2\frac{\sigma^2}{N}M\frac{\log N}{N}$ by the same reason as before. From Figure 3, we can see that our bounds precisely describe the decay of the test error. Our bounds are not 'bounding' the test errors in smaller instances due to the absence of the residue terms, which increases the interval of confidence of our approximation. But for larger instances, say $N > 100$, all upper and lower bounds, and the averaged test error converge to the same limit.

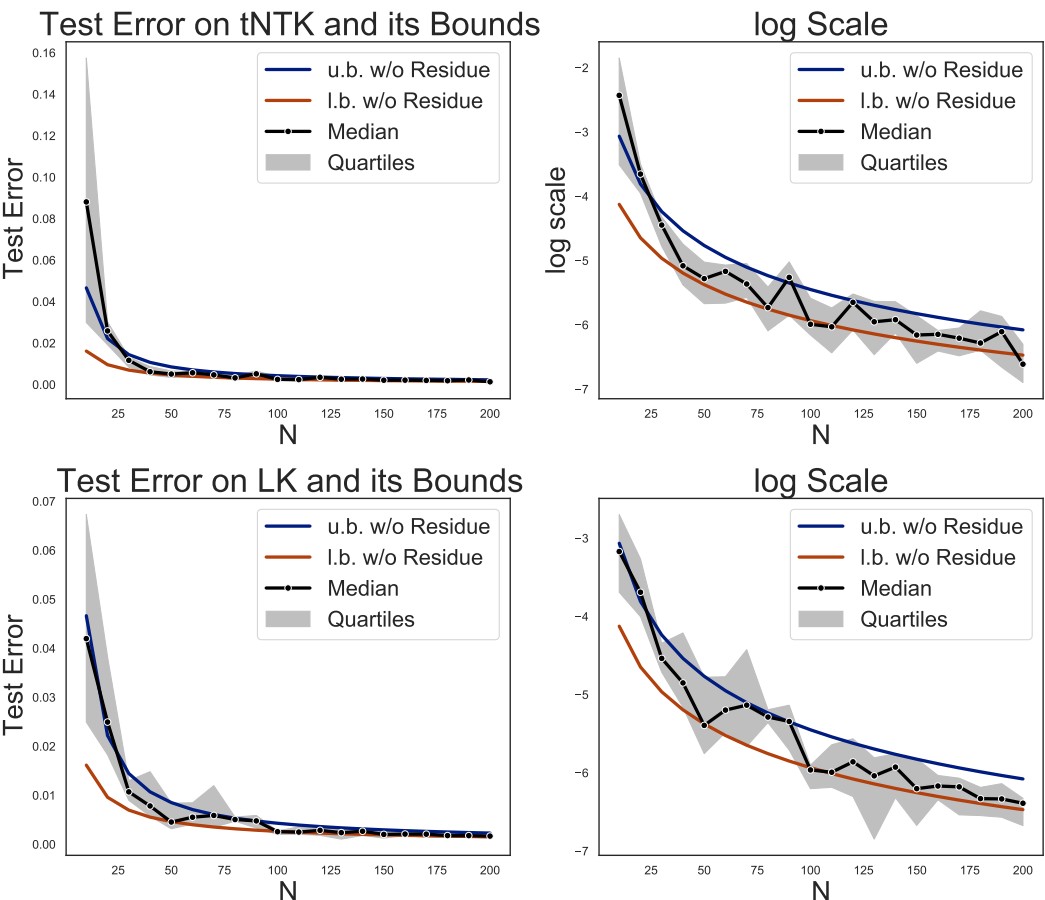

Figure 9: Our bounds comparing to the averaged test error with varying $N$, over 10 iterations. Same as Figure 3.

