# OpenReview forum: "A Theoretical Analysis of the Test Error of Finite-Rank Kernel Ridge Regression"
_NeurIPS.cc/2023/Conference — NeurIPS 2023 poster_

### Official Review · Reviewer_BvqB · 2023-06-14

**Soundness:** 4 excellent
**Presentation:** 4 excellent
**Contribution:** 3 good
**Rating:** 6
**Confidence:** 2

**Summary:**

The authors address the problem of kernel ridge regression with a finite rank kernel, in the under-paramatrized regime. They prove sharper upper bounds on the test error, compared to previous existing works.

**Strengths:**

The discussion is very well-written and easy to follow, with a number of illustrations being provided to set up the problem. Careful and detailed comparison to previous work is given, making it easy to understand the novelty of the present manuscript. Overall, the addressed problem is an important one, and I believe the derived bound will be of interest to the community. I have not checked the proof, and therefore give a low confidence score.

**Weaknesses:**

I do not have any major concern. A minor remark is that while all experimental details are provided in Appendix D, it would be good to add at least a short description of the target functions and regularization used in Fig.1 and 2 in the main text or in the corresponding captions.

**Questions:**

- (minor) While the comparison with the bound of [6] in Fig. 2 is compelling, is there a reason why the bound of [30] is not also plotted? Its addition, or a brief sentence justifying its non-inclusion, would be welcome.

---

> ### Author Rebuttal · Authors · 2023-08-06
>
> ## Experimental details
>
> **\[review\]** *” A minor remark is that while all experimental details are provided in Appendix D, it would be good to add at least a short description of the target functions and regularization used in Fig.1 and 2 in the main text or in the corresponding captions.”*
>
> **\[answer\]** Thank you for your suggestion; we will illustrate the experiments in more detail in the revised edition.
>
> ## Not plotting Rademacher bound
>
> **\[review\]** *”While the comparison with the bound of [6] in Fig. 2 is compelling, is there a reason why the bound of [30] is not also plotted? ”*
>
> **\[answer\]** The Rademacher bound is independent of $\lambda$, hence if we pick a very small $\lambda$, its learning curve as a function of $N$ would be way higher than ours and Bach’s. We will indeed add a sentence about this in the revised version, thank you for pointing this out.

---

> > ### Comment · Reviewer_BvqB · 2023-08-11
> > **Acknowledgement**
> >
> > I thank the authors for the explanations. I wish to stand by my original score, and stress once again that my assessment is limited by my inability of checking the proof, whence low confidence.

---

> > > ### Author Response · Authors · 2023-08-13
> > >
> > > Thank you for your positive rating and brief comments on our paper. We greatly appreciate your time and consideration.

---

### Official Review · Reviewer_S6gY · 2023-07-05

**Soundness:** 3 good
**Presentation:** 2 fair
**Contribution:** 3 good
**Rating:** 6
**Confidence:** 3

**Summary:**

The paper highlights the inadequacy of existing statistical learning guarantees for general kernel regressors when applied to finite-rank kernels. The authors have addressed this issue by developing non-asymptotic upper and lower bounds for the test error of finite-rank kernel ridge regression. These new bounds are more precise than the previously established ones and are applicable for any regularization parameters. This research provides a more dependable framework for utilizing finite-rank kernels in machine learning applications.

**Strengths:**

1. The paper addresses an important gap in the current understanding of machine learning algorithms by developing more accurate and reliable bounds for finite-rank kernel ridge regression, which is frequently used in practical applications.

2. The research provides a more precise and dependable framework for using finite-rank kernels in machine learning problems, which could result in better performance and more efficient algorithms.

**Weaknesses:**

1. The paper only considers under-parameterized regime.

2. Low bound is a main argument of this paper, but all details are given in the Appendix.


**Questions:**

1. It looks a little bit weird that you don't need to take $\lambda=O(N^\theta)$ for some $\theta<0$ to balance the bias and variance. Any insights for this?

2. In Theorem 4.1 and Corollary 4.3.1, what are the mild conditions on the kernel K? I cannot find the mild condition on input distribution either.

**Limitations:**

The authors have adequately addressed the limitations.

---

> ### Author Rebuttal · Authors · 2023-08-06
>
> ## Lower bound
> **\[review\]** *”Low bound is a main argument of this paper, but all details are given in the Appendix.”*
>
> **\[answer\]** Yes, we should have stated the lower bound in the main theorem explicitly. As mentioned in responses to other reviewers, it is in hindsight very clear to us that the lower bound should have been duly highlighted in the main text; we will do so in the revision
>
> ## Dependence of $\lambda$ w.r.t. $N$
>
> **\[review\]** *”It looks a little bit weird that you don't need to take $\lambda=O(N^\theta)$
>  for some $\theta<0$...?”*
>
> **\[answer\]** Indeed, this is one of the key contributions of our work. Compared with [Rudi et al. (2015)](https://papers.nips.cc/paper_files/paper/2015/hash/03e0704b5690a2dee1861dc3ad3316c9-Abstract.html), [Rudi and Rosasco (2017)](https://proceedings.neurips.cc/paper_files/paper/2017/file/61b1fb3f59e28c67f3925f3c79be81a1-Paper.pdf) and [Bach (2023)](https://www.di.ens.fr/~fbach/ltfp_book.pdf), in our theorem, the regularization $\lambda$ can be chosen independent of sample size $N$ or kernel rank $M$.
>
> ## Balancing bias and variance
>
> **\[review\]** *”It looks a little bit weird that you don't need to take $\lambda=O(N^\theta)$
>  for some $\theta<0$ to balance the bias and variance. Any insights for this?”*
>
> **\[answer\]** For simplicity in the main theorem, we have suppressed the dependency of $\lambda$ in variance by upper bounding the term $S=\sum_{k=1}^M\frac{\lambda\_k^2}{(\lambda\_k+\lambda)^2}$ by $M$. Please see Proposition C.14 for a more detailed approximation of the variance. Hence it is possible to derive an optimal $\lambda>0$ to balance the bias-variance tradeoff which depends on the kernel spectrum. Since we have the lower bound as well, such $\lambda$ would be minimax optimal. Traditional analysis suggests $\lambda$ should be in the magnitude of $\sqrt{\sigma}$, but our detailed analysis could also relate the choice $\lambda$ with the spectrum $\lambda_k$. We will put this as a possible future research direction.
>
> ## Mild condition
>
> **\[review\]** *”In Theorem 4.1 and Corollary 4.3.1, what are the mild conditions on the kernel K? ”*
>
> **\[answer\]** The mild condition on the kernel $K$ is indeed Assumption C.15, stating for $x\sim\rho$, the distribution $\psi\_k(x)$ is sub-Gaussian with sub-Gaussian norm bounded from above, where $\psi\_k$ is any eigenfunction of $K$. Note that for compact input space $\mathcal{X}$ and finite rank $M$, Assumption C.15 always holds, since any bounded distribution is sub-Gaussian. This assumption is often used in KRR results, for example, in [Tsigler and Bartlett (2023)](https://www.jmlr.org/papers/v24/22-1398.html).

---

> > ### Comment · Reviewer_S6gY · 2023-08-18
> > **Acknowledgement**
> >
> > Thank the authors for addressing my concerns. I keep my score unchanged.

---

### Official Review · Reviewer_cigT · 2023-07-05

**Soundness:** 3 good
**Presentation:** 2 fair
**Contribution:** 3 good
**Rating:** 7
**Confidence:** 3

**Summary:**

This paper analyzes the test error of ridge regression with a finite rank kernel. The finite rank kernel appears in several practical settings such as transfer learning and random neural networks. The authors provide a new analysis in this setting using tools from random matrix theory. A detailed comparison to other generalization bounds is presented.

**Strengths:**

1. New generalization results in a practical and challenging setting.
2. New analysis techniques.


**Weaknesses:**

1. The comparison to standard generalization results is not clear (Eq. (12)). Both bounds scale as $\sqrt{\frac{\log(n)}{n}}$. It is not clear which one is tighter.
2. The technical details of the proof are not given and the novelty of the analysis is not explained in detail. Instead, most of the paper is devoted to a discussion and experiments.
3. There are some unclear technical issues (see questions below).


**Questions:**

1. The bound in Theorem 4.1 and the bound in Eq. (12) both scale as $\sqrt{\frac{\log(n)}{n}}$. Which one is tighter and why? What is C in Eq. (12)?
2. The test error definition is not standard: Shouldn’t the label be $y$ in the MSE and not $\tilde{f}$? Is the definition of the test error the same in other works? Also, the bias-variance decomposition does not seem to be standard. Usually the learned predictor is the same in the bias expression (e.g., see the Wikipedia definition of bias-variance decomposition), whereas here the predictor is different: it changes from $f_{Z,\lambda}$ to $f_{(X,f(X)),\lambda)}$. Is this standard for analyzing kernel methods?
3. Where in the analysis is the under-parameterized assumption ($M <N$) needed?
4. In Eq. (10) if $\tilde{\gamma}_{>M} = 0$, the bias is exactly zero for all $N$? Is this a mistake?



**Limitations:**

Limitations are not discussed in detail. Some of the limitations are mentioned throughout the paper. Maybe it will be helpful to add a limitations section and summarize all limitations.

---

> ### Author Rebuttal · Authors · 2023-08-06
>
> ## Technical details and novelty
>
> **\[review\]** *”The technical details of the proof are not given and the novelty of the analysis is not explained in detail.”*
>
> **\[answer\]** We will refactor the revised paper so that these things are stated more clearly and more prominently, while relegating some of the numerics to the appendices. Due to space constraints in the paper, we apologize for the absence of a paragraph dedicated to technical details. In short, the main technical tool we used in the paper is the concentration of the spectrum of a centered random matrix with sub-Gaussian entries followed by the Neumann expansion of a matrix inverse. Another novel aspect is that we demonstrate how this same novel technique can also be used to derive the lower bound for the test error.
>
> ## Rademacher bound in Equation (12)
>
> **\[review\]** *”The bound in Theorem 4.1 and the bound in Eq. (12) both scale as
>  $\frac{\log N}{N}$. Which one is tighter and why? What is C in Eq. (12)?”*
>
> **\[answer\]**
> Here is Theorem 10.7 from [Mohri et al. (2018)](https://cs.nyu.edu/~mohri/mlbook/):
> > Let $K:\mathcal{X}\times\mathcal{X}\to\mathbb{R}$ be a PDS kernel, $\mathbf{\Phi}:\mathcal{X}\to\mathcal{H}$ a feature mapping associated to $K$, and $H=\\{ x\mapsto\mathbf{w}\cdot\mathbf{\Phi}:\\|\mathbf{w}\\|\_\mathcal{H}\leq\Lambda \\}$. Assume there exists $r\>0$ such that $K(x,x)\leq r$ and $\|f(x)\|\leq \Lambda r$ for all $x\in\mathcal{X}$. Then for any $\delta>0$, with probability at least $1-\delta$, the following holds for all $h\in H$:
> $$ R(h) \leq \hat{R}(h) + \frac{8r^2\Lambda^2}{\sqrt{N}}\Big(1+\frac{1}{2}\sqrt{\frac{\log \frac{1}{\delta}}{2}}\Big), $$
> > where $R(h)=\mathbb{E}_x[(h(x)-\tilde{f}(x))^2]$ is the test error and $\hat{R}(h)=\frac{1}{N}\sum\_{i=1}^N[(h(x\_i)-\tilde{f}(x\_i))^2]$ is the train error.
>
> Our bound in Theorem 4.1 is tighter than equation (12) in two ways: first, the decay rate of bias bound in equation (10) is $\frac{\log N}{N}$ as $\lambda\to0$, which one cannot obtain in Rademacher bound, since it is insensitive to $\lambda$; second, the total decay in the variance bound in equation (9) is $\frac{1}{N}\cdot\sqrt{\frac{\log N}{N}}$.
>
> ## Test error and bias-variance decomposition
>
> **\[review\]** *”The test error definition is not standard… Is the definition of the test error the same in other works? Also, the bias-variance decomposition does not seem to be standar…Is this standard for analyzing kernel methods?”*
>
> **\[answer\]** Our definition of test error measures the difference between the regressor trained on a noisy sample and the (noiseless) target function, which is standard in KRR research, and is equivalent to, for example, the generalization error in [Mohri et al. (2018)](https://cs.nyu.edu/~mohri/mlbook/) , the integrated square risk in [Liang (2019)](https://arxiv.org/abs/1808.00387), out-of-sample risk in [Hastie et al. (2020)](https://arxiv.org/abs/1903.08560), excess risk in [Bach (2023)](https://www.di.ens.fr/~fbach/ltfp_book.pdf). Previous works and we choose this definition as the noise only comes from the noisy sample (train set), but not from the test point. Indeed, we have
> $$
> \mathbb{E}\_{x,y} [(f\_{\mathbf{Z},\lambda}(x)-y)^2]
> = \mathbb{E}\_{x,\epsilon}[(f\_{\mathbf{Z},\lambda}(x)-\tilde{f}(x)-\epsilon)^2]
> = \mathbb{E}\_{x}[(f\_{\mathbf{Z},\lambda}(x)-\tilde{f}(x))^2] - 2\mathbb{E}\_{x,\epsilon}[\epsilon(f\_{\mathbf{Z},\lambda}(x)-\tilde{f}(x))] +\mathbb{E}\_{x,\epsilon}[\epsilon^2]
> $$
> $$
> = \mathbb{E}\_{x}[(f_{\mathbf{Z},\lambda}(x)-\tilde{f}(x))^2] + \sigma^2
> $$
> since $\epsilon$ is noise with mean zero and variance $\sigma^2$ independent to $x$.
> Note that we are averaging over all test points $x$ but fix the sample $\mathbf{Z}$ (as done in prior work, e.g. [Bach (2023)](https://www.di.ens.fr/~fbach/ltfp_book.pdf)), and our main theorem gives high probability bound over the sampling $\mathbf{Z}$ on the average mean square distance between the regressor and the target function. Also, our bias-variance decomposition is also standard and performed the same as in the above cited works, where we separate the effect of the sample noise to the test error as the variance. we will add a remark in the revision.
>
> ## Under-parametrization
>
> **\[review\]** *”Where in the analysis is the under-parameterized assumption ($M<N$) needed?”*
>
> **\[answer\]** The requirement that $N>M$ ensures the fluctuation matrix $\mathbf{\Delta}\in\mathbb{R}^{M\times M}$ is full-rank with operator norm that converges to zero as $N\to\infty$, then we argue with Neumann series expansion. Please see Section C.2 in the appendix for details.
>
> ## Consistent case ($\tilde{\gamma}_{ > M}=0$)
>
> **\[review\]** *”In Eq. (10) if $\tilde{\gamma}\_{ > M}=0 $, the bias is exactly zero for all $N$? Is this a mistake?”*
>
> **\[answer\]** Note that for $\tilde{\gamma}\_{ > M} = 0 $, we have the target function $\tilde{f}$ belongs to the finite-dimensional RKHS $\mathcal{H}$. Indeed, since the bias measures the fitness of the regressor with a noiseless sample $(\mathbf{X},\tilde{f}(\mathbf{X}))$, and if there is no regularization, we have the regressor
> \begin{equation*}
> 	f_{(\mathbf{X},\tilde{f}(\mathbf{X})),\lambda} = \arg \min\_{f\in\mathcal{H}}
> \sum_{i=1}^N(f(x)-\tilde{f}(x))^2
> \end{equation*}
> and it is clearly equal to $\tilde{f}$, hence the bias $\mathbb{E}\_x[(f\_{(\mathbf{X},\tilde{f}(\mathbf{X})),\lambda}-\tilde{f})^2]=0$ and this is no mistake.

---

> > ### Comment · Reviewer_cigT · 2023-08-12
> >
> > Thanks for the response. The authors addressed my concerns. I am raising the score.

---

> > > ### Author Response · Authors · 2023-08-13
> > >
> > > We sincerely appreciate your positive feedback and rating on our paper. Your insightful comments have immensely contributed to the improvement of our work. We are grateful for your time and expertise, and we are thrilled that our paper resonated positively with you.
> > > Once again, thank you for your valuable contribution to the refinement of our paper.

---

### Official Review · Reviewer_Gn3m · 2023-07-08

**Soundness:** 2 fair
**Presentation:** 2 fair
**Contribution:** 1 poor
**Rating:** 4
**Confidence:** 4

**Summary:**

The paper studies the kernel ridge regression under the non-asymptotic setting. The authors give the upper and lower bounds for bias and variance term, respectively. The authors argue the results improve upon those in Bach 2023.

**Strengths:**

The paper is well-structured. The authors give rigorous proofs, following by careful experiments.

**Weaknesses:**

1. The paper requires further improvement and polishing in writing. To name a few, line 57-58; line 107-110.

2. The paper would benefit from a more consistent and standardized use of symbols and notations. For example, in line 90, it would be better to use L_2(\rho) instead of L_\rho^2. In lien 92, the notation of \tilde{f}(\textbf{X}) is not proper. In Definition 3.1, it would be better to include the decreasing order of eigenvalues in the statement rather than adding an additional remark 3.2. In line 200-207, notation K^{(\infty)}

3. As mentioned in Bach 2023, more refined bounds can be found in Rudi et al. 2015, Rudi and Rosasco 2017. However, the authors failed to mention them and other related results in the comparison. In the absence of such comparisons, it is hard to tell the novelty and improvements of the current submission.

4. The dependency of \lambda seems to be incorrect for the variance term.

5. Given Corollary 4.3.1 in the submission, I cannot see significant improvements against those in Bach 2023. Also, the authors did not give a proper explanation for considering \lambda goes to 0.

**Questions:**

Could the authors mention the lower bound for the problem?

With optimal choice of \lambda, we can derive the optimal upper bound for the test error. However, the current result does not show a bias-variance tradeoff with respect to \lambda. Could the authors explain?

**Limitations:**

No.

---

> ### Author Rebuttal · Authors · 2023-08-04
>
> ## Polishing
> **\[review\]** *"The paper requires further improvement and polishing in writing."*
>
> **\[answer\]** Thank you for pointing out certain issues with the writing; we will address them in the revised edition.
>
> ## More references
>
> **\[review\]** *"As mentioned in Bach 2023, more refined bounds can be found in Rudi et al. 2015, Rudi and Rosasco 2017. However, the authors failed to mention them... it is hard to tell the novelty and improvements of the current submission."*
>
> **\[answer\]** We agree with the comment made by the reviewer and will discuss both papers  [Rudi et al. (2015)](https://papers.nips.cc/paper_files/paper/2015/hash/03e0704b5690a2dee1861dc3ad3316c9-Abstract.html), [Rudi and Rosasco (2017)](https://proceedings.neurips.cc/paper_files/paper/2017/file/61b1fb3f59e28c67f3925f3c79be81a1-Paper.pdf)  as related work in the revised edition.The citations were not included in the current submission because their focus is on the upper bound of the test error for specific sketching algorithms (Nyström approximation and random feature) while our work focuses on any under-parametrized finite rank kernel ridge regression. One main contribution is that we provide both tighter upper bound and tighter lower bound than the two (they derive the same convergence rate on the upper bound up to constants but they do not derive a lower bound). Another major difference is that our bound works for any regularization $\lambda$, while the mentioned prior works require it to be dependent on the sample size $N$ (we for instance refer the reviewer to Theorem 1 in  [Rudi and Rosasco (2017)](https://arxiv.org/abs/1602.04474)).
>
> ## Dependence on $\lambda$
>
> **\[review\]** *"The dependency of $\lambda$ seems to be incorrect for the variance term."*
>
> **\[answer\]**
> For simplicity in the main theorem, we have suppressed the dependence of $\lambda$ in the variance by upper bounding the term $S=\sum_{k=1}^M\frac{\lambda_k^2}{(\lambda_k+\lambda)^2}$ by M. Please see Proposition C.14 for a more detailed bound on the variance. Indeed this sum $S$ is less than the so-called effective dimension $\mathcal{N}_M(\lambda) =\sum \frac{\lambda_k}{\lambda_k+\lambda}$, which appears in the upper bound of the variance in Lemma 6 in [Rudi and Rosasco (2017)](https://arxiv.org/abs/1602.04474)). Ultimately, our bound is sharper.
>
> ## Contribution of Corollary 4.3.1
>
> **\[review\]** *"Given Corollary 4.3.1 in the submission, I cannot see significant improvements against those in Bach 2023."*
>
> **\[answer\]** The three major improvements are: a lower bound (that matches the upper bound up to constants), a better dependence on $\lambda$ (note that prior bound are vacuous for small $\lambda$, see details above) and tighter leading coefficients. Please also see our experimental results (Figure 2) that clearly shows significant improvements of our bounds.
>
> ## Ridgeless regression
>
> **\[review\]** *"Also, the authors did not give a proper explanation for considering $\lambda$ goes to 0."*
>
> **\[answer\]** Stating the case where $\lambda\to0$ is a way to understand the role of regularization. It also gives us some insights into the behavior of the bound for small (but non-zero) values of $\lambda$, for which our bound is also tighter than prior work. In practical scenarios, excessive l2-regularization can adversely affect test error, making smaller values of $\lambda$ equally noteworthy.
>
> ## Lower bound
>
> **\[review\]** *"Could the authors mention the lower bound for the problem?"*
>
> **\[answer\]** Yes, we have addressed the lower bounds in Theorems C.20-21 in the appendix and this is to our knowledge novel. We will do as you suggest and explicitly state them in the main theorem.
>
> ## optimal choice of $\lambda$
>
> **\[review\]** *"With optimal choice of $\lambda$, we can derive the optimal upper bound for the test error. However, the current result does not show a bias-variance tradeoff with respect to $\lambda$. Could the authors explain?"*
>
> **\[answer\]** From the previous paragraphs, it is possible to derive an optimal $\lambda>0$ to balance the bias-variance tradeoff which depends on the kernel spectrum. We will put this as a possible future research direction.

---

> > ### Comment · Reviewer_Gn3m · 2023-08-12
> >
> > Thank the authors for the detailed response!
> >
> > However, it seems the authors misunderstood the lower bound mentioned in my review. I was not talking about the lower bounds for these "upper bounds". The lower bound that I referred to is that for the estimation problem introduced in Sec 3.1, which can help understand whether the upper bounds are optimal in the minimax sense. I also cannot see the point why the authors consider these "lower bounds" in Thm C.20, C.21. In general the lower bound that I refer to can tell us how hard the estimation problem is no matter what estimation methods are employed. The "lower bounds" in Thm C.20, C.21 can only tell us how tight the analyses of the upper bounds in the subscription are using the kernel ridge regressor.
> >
> > Also, as agreed by the authors, the papers by Rudi and the coauthors derived the same convergence rate on the upper bound up to constants.
> >
> > Given these facts, I would like to keep my original score.

---

> > > ### Author Response · Authors · 2023-08-13
> > >
> > > Dear reviewer,
> > >
> > > We extend our sincere gratitude for your valuable feedback on our manuscript and for engaging in a discussion with us.
> > >
> > > ## Minimax rate
> > >
> > > It appears that your inquiry pertains to the minimax rate, as described, for instance, in [Haghifam et al.  (2023)](https://proceedings.mlr.press/v201/haghifam23a/haghifam23a.pdf). Is this indeed what you are referring to?
> > >
> > > While we concur that the minimax rate is a noteworthy concern, we wish to elucidate that the primary focus of our paper diverges from this particular aspect. For instance, the rates you mentioned are independent of the algorithm, and our focus lies in KRR. We are of the opinion that our exposition is clear and devoid of any ambiguity pertaining to this matter. Our paper undertakes the specific task of bounding the bias-variance of Kernel Ridge Regression (KRR), which is in itself an important topic in machine learning (other reviewers seem to agree on this). We would be pleased to incorporate a detailed discussion and include relevant citations if the reviewer could be more specific in their request.
> > >
> > > In case we have again misunderstood your feedback, we would be more than delighted to engage in further discussion on this matter.
> > >
> > > ## Rudi et al.
> > >
> > > A pivotal differentiating aspect of our work, in comparison to prior endeavors, is the universality of our proposed bound, which remains applicable across the entire spectrum of $\lambda$ values. This stands in stark contrast to the approach adopted by [Rudi and Rosasco (2017)](https://arxiv.org/abs/1602.04474), where $\lambda$ has to scale in proportion to $1/N$. Another aspect to consider is that [Rudi and Rosasco (2017)](https://arxiv.org/abs/1602.04474) do not establish a lower bound, a topic we delve into further in the subsequent explanation.
> > >
> > > ## Why do we care about the lower bound?
> > >
> > > This element serves to elucidate the precision of the upper bound for KRR, an aspect we believe constitutes a noteworthy contribution. It is pertinent to note that preceding works have primarily focused on upper bounds, making our emphasis on a comprehensive analysis a noteworthy contribution.
> > > The importance of the lower bound is also discussed among other researchers, like [Tsigler and Bartlett (2023)](https://www.jmlr.org/papers/v24/22-1398.html).
> > >
> > >
> > > Once again, we express our sincere appreciation for your time and attention; and we hope that you might reconsider your final rating.

---

### Official Review · Reviewer_MpJJ · 2023-07-27

**Soundness:** 4 excellent
**Presentation:** 3 good
**Contribution:** 3 good
**Rating:** 7
**Confidence:** 4

**Summary:**

This paper gives a refined analysis of the test error of kernel ridge regression with a finite kernel in the underparametrized regime, where the rank M of the kernel is smaller than the number of data points N. The analysis improves upon previous ones by providing tighter non-asymptotic upper and lower bounds, which remain tight even as the ridge parameter goes to zero. When the target function lies in the RKHS, the bounds show that estimation is consistent as N goes to infinity.

**Strengths:**

The paper is well written and clear. The authors state the contributions clearly and compare carefully with previous work. Establishing tight bounds for this basic learning setting is an important problem.

**Weaknesses:**

No major weaknesses, just some minor typos. I would also suggest that the authors include the explicit lower bounds in the main paper. Given that this a major focus of the paper it should not be required for the reader to go to the appendix. It would also be nice for the appendix to give a bit clearer account for the technical differences in the proof techniques of their bounds versus previous ones.

**Questions:**

- Can the authors elaborate the technical difficulties in extending beyond the finite rank setting, or even when M ~ N? Can analogous results be shown when the kernel is approximately low rank?

- It is unclear that realistic NTK or random features will fall in this underparametrized regime. Can the authors give elaborate if this should be the case? Is it possible to experiment in more realistic NTK settings to truncate the kernel and apply the bounds?

**Limitations:**

No limitations. Authors give some directions or future work.

---

> ### Author Rebuttal · Authors · 2023-08-04
>
> ## Explicit lower bound
>
> **\[review\]** *“I would also suggest that the authors include the explicit lower bounds in the main paper. Given that this a major focus of the paper it should not be required for the reader to go to the appendix.”*
>
> **\[answer\]**  We concur with the reviewer's viewpoint and, consequently, plan to relocate the lower bound in the main theorem and move the proof technique section to the appendix.
>
> ## Technical tool
>
> **\[review\]** *“It would also be nice for the appendix to give a bit clearer account for the technical differences in the proof techniques of their bounds versus previous ones.”*
>
> **\[answer\]** The main technical tools we used in the paper are 1) more careful algebraic manipulations when dealing with terms involving the regularizer $\lambda$ and 2) a concentration of the spectrum of a centered random matrix with sub-Gaussian entries followed by the Neumann series expansion of a matrix inverse. To this end, we require that $M<N$, such that the fluctuation matrix $\mathbf{\Delta}\in\mathbb{R}^{M\times M}$ is of full rank with operator norm converges to zero as $N\to\infty$, then we argue with Neumann series expansion in $\mathbf{\Delta}$, instead of in the terms with leading factor $\lambda$. Hence, unlike previous work, our result holds for any $\lambda>0$ which can be chosen independence to $N$. See Section C.2 in the appendix for details.
>
> ## Under-parametrized regime
>
> **\[review\]** *"Can the authors elaborate the technical difficulties in extending beyond the finite rank setting, or even when M ~ N?"*
>
> **\[answer\]** In the main theorem, we require $N$ to be exponential in $M$ in order to have a decay of the order $\frac{\log N}{N}$. But as indicated in the remark of Lemma C.17 (lines 547 - 551) in the appendix, we can relax the requirement that N to be only polynomial in M, with the tradeoff of a slower decay rate. For the case $M=N$, it is where the well-known variance explosion would happen unless we regularize, that is, unless $\lambda > 0$. Since this paper focuses on bounding the test error with a general $\lambda$, we refer the readers to [Bach (2023)](https://www.di.ens.fr/~fbach/ltfp_book.pdf) for the test error bound for $M\geq N,\ \lambda > c$. For over-parametrized regime ($M>N$) where the kernel is approximately low-rank, our strategy does not does not apply, but it is one of our future research direction. Note that there are already some published non-asymptotic result, for instance, see  [Tsigler and Bartlett (2023)](https://www.jmlr.org/papers/v24/22-1398.html).
>
> ## Practical use
>
> **\[review\]** *"It is unclear that realistic NTK or random features will fall in this underparametrized regime. Can the authors give elaborate if this should be the case? Is it possible to experiment in more realistic NTK settings to truncate the kernel and apply the bounds?"*
>
> **\[answer\]** The NTK of any finite-width (shallow) fully-connected network is finite-dimensional with random eigenfunctions. For simplicity, we truncate the NTK (of a infinite-width network) by selecting only the first few eigenfunctions in the experiment as a toy example. In general, as long as the sample size $N$ is larger than the number of trainable parameters in the network, our result can be applied. In practical scenarios like fine-tuning or transfer learning in large language models, most of the pre-trained parameters are fixed except the last layer (see Section 2) or smaller low-rank matrices as in the popular LoRA approach [Hu et al.(2021)](https://arxiv.org/pdf/2106.09685.pdf). In this case, the training is simply a kernel ridge regression in the under-parametrized regime. We regard further analysis of such specific model trainings as a future research dierection.

---

> > ### Comment · Reviewer_MpJJ · 2023-08-17
> >
> > Thanks to the authors for providing clarifications. I will keep my original score.

---

### Author Rebuttal · Authors · 2023-08-06

**General response to all the reviewers:** Thank you for your comprehensive review of our paper. We sincerely appreciate the time and effort you have dedicated to evaluating our work.

Below, we outline the primary issues highlighted by the reviewers based on our interpretation of their feedback. We would be grateful if the reviewers could indicate if they have any additional inquiries or if they find our responses satisfactory.

### Lower bound of the test error
We acknowledge your valuable feedback regarding the need for emphasizing the lower bound of the test error in the main text. This aspect represents a significant and novel contribution of our research since we are not aware of other prior work deriving a lower bound that matches the upper bound. We will ensure that it receives proper attention in the revised version.

### Dependence of $\lambda$
In our paper, the regularization $\lambda$ can be chosen to be any positive number independent of $N$, in contrast to previous works (please see individual rebuttals for more details). This is also a novel contribution of our research.

### Technical details
Additionally, we understand the importance of providing more technical details in the main text to aid readers in understanding the overall proof procedure. We provide further details in the detailed answers and we will make the necessary improvements to address this concern.


We kindly request you to please refer to our detailed rebuttals to the individual reviews provided below. Once again, we sincerely thank you for your thoughtful evaluation and valuable suggestions.

---

### Decision · Program_Chairs · 2023-09-21

**Decision:**

Accept (poster)

**Comment:**

The paper elegantly contributes by analyzing kernel ridge regression in finite dimensions, offering sharp and novel insights. It's remarkable to find fresh results in this mature field, yet the paper effectively defines its scope and presents these findings clearly.